# Exploring the Role of *Staphylococcus aureus* in Inflammatory Diseases

**DOI:** 10.3390/toxins14070464

**Published:** 2022-07-06

**Authors:** Huanquan Chen, Junyan Zhang, Ying He, Zhuoyi Lv, Zhengtong Liang, Jianze Chen, Peishan Li, Jiawei Liu, Hongchen Yang, Ailin Tao, Xueting Liu

**Affiliations:** The Second Affiliated Hospital, The State Key Laboratory of Respiratory Disease, Guangdong Provincial Key Laboratory of Allergy & Clinical Immunology, Guangzhou Medical University, Guangzhou 510260, China; hq15767895062@163.com (H.C.); kuhn2000@163.com (J.Z.); heying0605@163.com (Y.H.); lzy2589022034@163.com (Z.L.); liangzhengtong0620@163.com (Z.L.); cjz479957503@163.com (J.C.); lpeishan2022@163.com (P.L.); zhongshanljw@163.com (J.L.); yhc09282022@163.com (H.Y.)

**Keywords:** *Staphylococcus aureus*, toxin, inflammatory cells, pyroptosis, apoptosis, necroptosis, autophagy

## Abstract

*Staphylococcus aureus* is a very common Gram-positive bacterium, and *S. aureus* infections play an extremely important role in a variety of diseases. This paper describes the types of virulence factors involved, the inflammatory cells activated, the process of host cell death, and the associated diseases caused by *S. aureus*. *S. aureus* can secrete a variety of enterotoxins and other toxins to trigger inflammatory responses and activate inflammatory cells, such as keratinocytes, helper T cells, innate lymphoid cells, macrophages, dendritic cells, mast cells, neutrophils, eosinophils, and basophils. Activated inflammatory cells can express various cytokines and induce an inflammatory response. *S. aureus* can also induce host cell death through pyroptosis, apoptosis, necroptosis, autophagy, etc. This article discusses *S. aureus* and MRSA (methicillin-resistant *S. aureus*) in atopic dermatitis, psoriasis, pulmonary cystic fibrosis, allergic asthma, food poisoning, sarcoidosis, multiple sclerosis, and osteomyelitis. Summarizing the pathogenic mechanism of *Staphylococcus aureus* provides a basis for the targeted treatment of *Staphylococcus aureus* infection.

## 1. Introduction

*Staphylococcus aureus* (*S. aureus*), a Gram-positive bacterium, is one of the most notorious human pathogens, causing illnesses ranging from mild skin and wound infections to fatal sepsis or multi-organ failure.

*S. aureus* secretes a variety of toxins and enzymes and some proteins and peptides that also have certain virulence effects [1]. Different virulence factors play different roles in different diseases. In addition to the above virulence components, *S. aureus* itself has virulence effects. The proliferation of *S. aureus* at the site of infection leads to a dysbiosis of the flora at the site of infection, which drives a range of diseases [2]. *S. aureus* has a virulence regulator, Agr (auxotrophic gene regulator), which is a population sensing system. Agr upregulates many toxins and virulence determinants when bacterial cell density reaches a certain threshold, leading to the exacerbation of disease [3,4].

Inflammatory cells play an important role in *S. aureus* infection. *S. aureus* infection and toxins can activate a variety of inflammatory cells, such as keratinocytes [5], helper T cells [6], innate lymphoid cells (ILCs) [7], macrophages [8], dendritic cells (DCs) [9], mast cells [10], neutrophils [11], eosinophils [12], and basophils [13], which release inflammatory factors that accumulate at the site of infection and cause an inflammatory response. During infection, *S. aureus* can also induce host cell death through programmed forms of cell death, such as pyroptosis [14], apoptosis [15], necroptosis [16], and autophagy [17].

*S. aureus* leads to a variety of different infections ranging in severity from mild to fatal. A distinctive feature of *S. aureus* or methicillin-resistant *S. aureus* (MRSA) is its considerable reservoir of virulence factors that can lead to atopic dermatitis (AD) [18], psoriasis [19], pulmonary cystic fibrosis (CF) [20], allergic asthma [21], pneumonia [22], food poisoning [23], chronic granulomatous disease (CGD) [24], osteomyelitis [25], diabetic foot infections (DFIs), and many other diseases.

In this article, we focus on the types of immune cells and cell death mechanisms activated by *S. aureus* in various human diseases. All the abbreviations are mentioned in Table 1. We wrote this article to explain the relationship between *S. aureus* and related diseases and to describe the mechanism of action of *S. aureus* in related diseases. We hope it will provide some help in the treatment of related diseases.

## 2. Virulence Factors of *S. aureus*

The virulence factors of *S. aureus* can be divided into the following categories: (1) secreted virulence factors, including toxins and superantigens, the main function of which is to disrupt host cell membranes and induce target cell lysis and inflammation [26]; (2) extracellular enzymes, the main function of which is to break down host molecules for nutrition, promote bacterial survival and dissemination, etc. [26]; (3) surface proteins of *S. aureus*, whose main functions are adhesion, invasion, and immune escape [27], and (4) pathogen-associated molecular patterns (PAMPs), which promote inflammatory responses [28].

### 2.1. Secreted Virulence Factors

*S. aureus* can secrete a variety of enzymes and virulence factors that affect the immune system, leading to immune system dysregulation and the proliferation of auto-reactive T cells, as well as the development or progression of chronic autoimmune diseases. Virulence factors of *S. aureus* include pore-forming toxins (PFTs) [29], phenol-soluble modulins (PSMs) [30], exfoliative toxins (ETs) [31], and superantigens (SAgs) [32] that activate different types of immune cells and cause several different inflammatory and infectious diseases.

#### 2.1.1. PFTs

PFTs are important virulence factors secreted by bacteria that lead to cell lysis by forming pore structures in eukaryotic cell membranes. PFTs exert their toxic effects mainly by altering the permeability of cell membranes, leading to cell death [29]. However, the disruption of cell permeability is often preceded by the release of cytokines and the activation of intracellular protein kinases. PFTs include α-hemolysin (Hla) [33], β-hemolysin (Hlb), γ-hemolysin (Hlg) [34,35], α-toxin [36], and Panton–Valentine leukocidin (PVL) [37].

Hla, Hlb, and Hlg

Haemolysin is a pore-forming toxin, also known as a membrane-disrupting toxin. Haemolysin is a substance that lyses red blood cells and releases hemoglobin, a sensitive, complementary fixed antibody that binds specifically to the antigen type of the red blood cell. This antibody can be produced by stimulation of the surface antigen and can cause red blood cells to lyse and release hemoglobin [38]. Haemolysin includes Hla, Hlb, and Hlg.

Hla can induce the formation of small pores in the cell membrane, leading to the rapid release of K^+^ ions as well as inducing the production of interleukin-1β (IL-1β), IL-6, and IL-8 [33,39,40,41]. Activated endothelial and epithelial cells induce the activation of caspase-1 and the production of NLRP3-inflammasome through the release of nitric oxide (NO), leading to extracellular Ca^2+^ influx, the production of pro-inflammatory cytokines, and the pyroptosis of monocytes. Ca^2+^ influx also activates caspase-12, activates caspase-3, and induces apoptosis. Hla also acts on Toll-like receptor (TLR) 3/4 and mediates necroptosis [41,42,43,44,45,46].

Hlb and Hlg can promote inflammasome formation containing procaspase-1, ASC, and NLRP3. Inflammasome activation promotes the release of IL-1β and IL-18. Activated caspase-1 can also cleave GSDMD to form the N-terminal cleavage product of GSDMD, and these together induce pyroptosis [34,35].

2.α-Toxin

α-Toxin is a lecithin enzyme that breaks down lecithin, which is an important component of cell membranes. Therefore, α-toxin can damage the cell membrane of many cells, causing hemolysis, tissue necrosis, and vascular endothelial cell damage, increasing vascular permeability, and causing edema. α-Toxin activates macrophages and induces the activation of Th1 via CXCL-10. Th1 cells express interferon-γ (IFN-γ) in AD. α-Toxin induces early phagosomes (Rab5, Rab22b) to form autophagosomes (ILC3, Rab7), which induces cellular autophagy [36]. α-Toxin also activates RIPK3, caspase-8, and caspase-12, mediating necroptosis and apoptosis. α-Toxin acts in AD, granulomatous polyangiitis (GPA), pneumonia, chronic sinusitis, sepsis, and other diseases [47,48,49,50,51,52,53].

3.PVL

PVL is a pore-forming toxin produced by *S. aureus* that causes leukocyte destruction [37]. PVL can induce the lysis of macrophages and neutrophils, leading to cell death [54,55]. PVL activates receptor-interacting serine threonine kinase 1 (RIPK1), RIPK3, and mixed-lineage kinase-like protein (MLKL) through tumor necrosis factor receptor 1 (TNFR1) and TLR3/4, forming a complex that leads to necroptosis. PVL can also lead to apoptosis. PVL also stimulates K^+^ efflux, NLRP3 inflammasome production, and caspase-1 activation, leading to pyroptosis [35,56,57,58,59,60,61]. PVL causes pneumonia through the above cell death mechanisms [58,62,63,64,65,66].

#### 2.1.2. PSMs

PSMs are a family of amphipathic alpha-helical peptides found in *Staphylococci* [30]. PSMs contribute to biofilm structure and the propagation of biofilm-associated infections. PSMs include PSMα, PSMβ, and PSMγ.

PSMα

PSMα can induce necroptosis in atopic dermatitis and pneumonia [5,67,68,69,70]. PSMα induces the release of IL-1α and IL-36α from keratinocytes and induces the release of the pro-inflammatory cytokine IL-17, which mediates the skin inflammatory response to cause *S. aureus* infection [5]. PSMα induces the activation of keratinocytes and neutrophils, resulting in a series of pro-inflammatory responses (including cytokine production, leukocyte activation, and neutrophil chemotaxis) [71,72].

PSMα also activates and phosphorylates MLKL, as well as increases the expression and secretion of lactate dehydrogenase. Through MLKL and lactate dehydrogenase, PSMα can induce neutrophil necroptosis. The necroptosis of cells is the main pathological manifestation of *S. aureus* pneumonia [68].

2.PSMβ

PSMβ has an important role in the formation of biofilms. PSMβ activates and induces neutrophil aggregation through formyl-peptide receptor 2 (FPR2), which induces an inflammatory response [73]. In addition to the role of surfactant, PSMβ can also lyse erythrocytes and destroy them. However, in addition to the common properties of PSM, the unique role of PSMβ in *S. aureus* itself has not yet been investigated [30].

3.PSMγ (δ Toxin)

The *S. aureus* δ toxin is a member of the PSM family. The δ toxin is cytolytic to neutrophils and erythrocytes.

The δ toxin is cytolytic to neutrophils and erythrocytes. The staphylococcal δ-toxin promotes allergic skin disease in mice by inducing mast cell degranulation [74]. The δ toxin is an enterotoxin that is cytotoxic and increases cellular cAMP expression to inhibit water absorption in the ileum by altering the concentration of Na^+^ and Cl^−^ in the mucosa, causing diarrhea [75].

#### 2.1.3. Proteases

ETs

ETs are extremely specific serine proteases secreted by *S. aureus*. ETs can play a role in AD [31]. ETs are the primary toxins that play a role in staphylococcal scalded skin syndrome (SSSS), an abscessing skin disease [76].

ETs specifically recognize and hydrolyze the cell adhesion molecule desmoglein 1 (Dsg1), causing the dissociation of keratinocytes in human and animal skin and promoting skin infection by *S. aureus*. Three distinct ET subtypes (ETA, ETB, and ETD) were identified in *S. aureus* [77].

2.Serine Protease-Like Proteins (Spls)

Spls play a role in T cells and result in asthma [78]. Spls recognize and hydrolyze desmoplastic proteins in the superficial skin layer, inducing skin peeling and blister formation [79,80,81,82,83]. Spls trigger IgE antibody responses in most asthmatics. Peripheral blood T cells produce Th 2 cytokines after Spls stimulation. Therefore, Spls are considered to be triggering allergens in the allergic airway response to *S. aureus* [78]. T cells of CF patients produced more TH2 cytokines after stimulation by Spls [84]. Increased IgE concentrations of Spls were detected in the serum of CF patients relative to healthy controls [84].

3.Staphopain B (SspB)

SspB is a human strain of *S. aureus* that secretes papain-like proteases, and SspB has bacterial virulence. SspB can activate macrophages and lead to apoptosis [85]. SspB may contribute to the recruitment of host cells, including immunomodulatory pDCs and/or macrophages, which contribute to the initiation and maintenance of a chronic inflammatory state by *S. aureus* [86].

#### 2.1.4. SAgs

SAgs are a class of antigenic substances composed of bacterial exotoxins and retroviral proteins. They bind to most T cells and provide signals for T cell activation. SAgs are highly efficient T-cell mitogens and manipulate the host immune system. They can directly activate T lymphocytes, triggering the release of a large number of pro-inflammatory cytokines, such as IFN-γ, IL-2, and tumor necrosis factor (TNF) [32,87,88]. SAgs include *Staphyloccucal* enterotoxins (SEs) and toxic shock syndrome toxin 1 (TSST-1).

SEs

SEs include SEA, SEB, SEC, SEG, SHE, SEI, SEM, SEO, and SEQ (Table 2). SEA and SEB can play a role in Th1, Th2, and Th22 cells and induce apoptosis. In nasal polyposis, SEB can induce IL-21 expression, and IL-21 also induces differentiation of Th17 [89,90]. SEA can lead to AD and asthma. SEB can cause AD, asthma, and chronic sinusitis. SEC can also lead to AD and cancer [91,92,93]. SEA serves as a potent stimulant of PBMCs and induces the release of large amounts of cytokines and chemokines through the Src, ERK, and STAT pathways [94]. SEB, SEG, SHE, SEI, SEM, SEO, and SEQ can also lead to food poisoning [95,96,97,98,99,100].

2.TSST-1

TSST-1 is a bacterial SAg produced and secreted by *S. aureus*. TSST-1 can activate CD4^+^ T cells to produce large amounts of cytokines and lead to a systemic toxic response. In AD, TSST-1 can lead to B lymphocytes and keratinocytes [13,50].

#### 2.1.5. Secreted Enzymes (Exoenzymes) and Effectors

In addition to toxins, *S. aureus* secretes many virulence factors with proenzymatic effects. These proenzymatic virulence factors can be broadly classified into two types: cofactors, which are used to activate host enzymes, and enzymes that lyse and destroy host cells and tissues. Secreted enzymes (exoenzymes) and cofactors act on different substances with different specific mechanisms of action, but their main function is to break down host cells and tissues to obtain nutrients for their growth, reproduction, and propagation [26]. Secreted enzymes and effectors include EsxA, EsxB, coagulase (Coa), nuclease (Nuc), and adenosine synthase (AdsA), and staphopain (SspB).

EsxA and EsxB

EsxA and EsxB are small acidic proteins secreted by the early-secretion antigen-6 secretion system (ESS) as potential T-cell antigens of *S. aureus*. ESS is the basic virulence factor of *S. aureus*. EsxA and EsxB mediate the release of *S. aureus* from host cells, which can cause apoptosis [101,102].

2.Coa

Coa is an enzyme produced by *S. aureus*. Coa has thrombospondin-like activity and coagulates plasma treated with citric or oxalic acid. Coa can induce coagulation with vascular hemophilia factor binding protein [103]. Coa can lead to apoptosis [104].

3.Nuc and AdsA

Nuc can disrupt biofilms by breaking down extracellular DNA (eDNA) as well as mediating the escape of *S. aureus* from the NET. The NET is an innate immune defense mechanism by means of which invading pathogens are removed [105,106]. Moreover, Nuc can degrade DNA in abscesses or NETs.

The degradation product, nucleotide monophosphate, can be a substrate for the synthesis of another adenylate, synthase A (AsdA). AdsA degrades DNA into deoxyadenosine, which induces the apoptosis of macrophages around NETs by caspase-3 activation. Thus, AsdA can promote the survival of *S. aureus* [107]. Nuc and AdsA can induce bacteraemia and nephrapostasis [108].

4.Extracellular Adhesion Protein (Eap)

Eap is the substance that mediates the adhesion of bacteria to host cells. In the early stages of infection, *S. aureus* adheres and colonizes through the expression of Eap, secondary to infection. Eap can act on T cells and lung epithelial cells. It can also work in psoriasis and CF [109].

### 2.2. Surface Proteins of S. aureus (Cell Wall-Anchored (CWA) Proteins)

Invasion of organs and tissues by *S. aureus* from the blood stream requires not only immune evasion but also adhesion. Biofilm formation is an important way for *S. aureus* to maintain infection. Adhesion, proliferation, and detachment are the main processes of biofilm formation.

Surface proteins of *S. aureus* constitute a range of virulence factors, known as CWA proteins. CWA proteins play a key role in the adhesion phase. The sorting signal of CWA proteins is responsible for covalently coupling proteins to peptidoglycan (PGN) [110]. There are 24 different CWA proteins on the surface of *S. aureus*, including microbial surface components recognizing adhesive matrix molecules (MSCRAMMs), the near iron transporter (NEAT) motif protein family, three-helical bundle motif protein A, G5-E repeat family, legume-lectin, and cadherin-like domain protein [111]. CWA proteins play an important role in extracellular matrix (ECM) adhesion, host cell invasion, and immune response evasion. Therefore, targeting CWA proteins with vaccines can counteract *S. aureus* infections [112].

#### 2.2.1. MSCRAMM

MSCRAMMs are a series of proteins with similar amino acid sequences and very similar structures and functions. MSCRAMMs consist of two folded subdomains similar to IgG, and the two folded subdomains are adjacent to each other. MSCRAMM plays an important role in tissue invasion during *S. aureus* infection, including enabling *S. aureus* to adhere to and invade host cells and tissues, evade host immune attack, and induce biofilm formation [110]. Therefore, targeting the MSCRAMM protein is an alternative immunotherapeutic direction for the therapy of *S. aureus* infections [113].

#### 2.2.2. Staphylococcal Protein A (SpA)

SPA is a protein that forms the cell wall of *S. aureus*. It was found that SpA can form a complex with human immunoglobulin, and the complex formed by the two can induce the necrosis of various immune cells in the body [114]. SpA can induce apoptosis and cause osteomyelitis [115,116,117,118].

The second immunoglobulin-binding protein (Sbi) belongs to SpA and consists of four triple-helix bundles arranged in tandem. Two of the triple-helix bundles bind similarly to those of SpA and IgG. Sbi non-covalently binds lipoteichoic acid (LTA), which then binds to the cell envelope and helps *S. aureus* to evade attack by the host immune system.

*S. aureus* Sbi can induce IL-33 production from keratinocytes in AD [119]. IL-33 can induce itch. IL-33 is a type 2 cytokine and a major regulator of chronic itch [120]. Sbi-inducible expression of IL-33 causes pruritus in AD [121]. It is therefore a virulence factor that promotes type 2 immune response.

### 2.3. PAMPs

PAMPs are bacterial-specific structures that typically activate TLRs, which act as neutrophil activators [28]. PAMPs of *S. aureus* include triacyl lipopetides, diacyl lipoproteins, and LTA.

#### 2.3.1. Triacyl Lipopetides and Diacyl Lipoproteins

Triacyl lipopetides and diacyl lipoproteins are components of the *Staphylococcal* cell wall that provide structural integrity and protection against virulence organisms from the host immune system. Triacyl lipopetides and diacyl lipoproteins can be distinguished by subtle differences in TLR 1 and TLR 6 interactions with TLR2 [122]. Triacyl lipopetides and diacyl lipoproteins can induce activation of the TLR 2 and NLRP3 inflammasome. NLRP3 activation induces pyroptosis. Lipopetides and diacyl lipoproteins can induce the activation of TLR 2 and the NLRP3 inflammasome [35,69,123]. Triacyl lipopetides and diacyl lipoproteins can make use of sepsis [69,124].

#### 2.3.2. LTA

LTA is an adhesion affinity agent associated with surface adhesion and a modulator of the bacteria’s own cell wall lysis enzymes (Muramidases). LTA is released from the bacteria after rupture and death. LTA can induce the phenomenon of passive immune killing. LTA promotes macrophage activation and the expression of IL-1β and IL-18 [125]. LTA activates neutrophils and macrophages and leads to pyroptosis. LTA can also work in food poisoning [23,125,126,127].

#### 2.3.3. PGN

PGN is the main structural component of the cell wall [128], a glycopolymer that maintains the shape of the bacteria. PGN is also a key factor in bacterial recognition by the host immune system [129].

PGN induces the activation of keratinocytes, Langerhans cells (LCs), mast cells, macrophages, CD4^+^T cells, Th1 cells, and microglia, and induces the release of several cytokines, including TNF-α, IL-1, and IL-4 [17,130,131,132,133,134,135,136,137,138,139]. PGN causes autophagy, which drives AD, sepsis, pneumonia, and psoriasis [17,130,131,132,133,140,141]

*S. aureus* contains large amounts of SpA, which evades phagocytosis and uptake by phagocytes by binding to immunoglobulin (Ig) molecules and then covering the bacterial surface. Peptidoglycan can promote the B-cell superantigen activity of SpA [142].

A summary of the above factors activating immune cell types and the types of induced cell death and induced diseases is shown in Table 3.

## 3. Different Inflammatory Cell Types Involved in *S. aureus* Pathogenesis

The cells involved in *S. aureus* infection and inflammation include keratinocytes, T cells (helper T cells, ILCs), macrophages, DCs, mast cells, neutrophils, eosinophils, and basophils.

### 3.1. Keratinocytes/Epithelial Cells

Keratinocytes are the main cellular components that make up the epidermis. *S. aureus* expresses PSMα, which is a group of secreted virulence peptides.

In AD, *S. aureus* accumulates in the lysosomes of keratinocytes and induces IL-1α secretion via TLR9 [152]. *S. aureus* expresses PSMα acting on keratinocytes, induces IL-1α and IL-36α release from keratinocytes via myd88 signaling, induces γδt cell and ILC3 production of IL-17, and promotes neutrophil infiltration [5]. In addition, *S. aureus* promotes IL-36α secretion from keratinocytes and promotes IgE production and allergic inflammatory response [70]. *S. aureus* promotes IL-1β expression by keratinocytes and induces skin regeneration through keratinocyte-dependent IL-1R-MyD88 signaling [153].

*S. aureus* Sbi promotes IL33 secretion by keratinocytes, and IL-33 promotes type 2 immune responses [119,154]. IL-33 also has an important role in pruritus [155]. *S. aureus* diacylated lipoprotein is a TLR2 and TLR6 ligand, which activates keratinocytes via TLR2-TLR6 heterodimers to produce TSLP, which promotes T(H)2-type inflammation and thus AD [156]. The keratinocyte-expressed TSLP acts directly on TRPA1-positive sensory neurons, triggering intense pruritus-induced scratching [157].

In nasal polyp tissue, *S. aureus* can directly induce the release of the epithelial cell-derived cytokines TSLP and IL-33 by binding to TLR 2, thereby potentially propagating the expression of type 2 cytokines IL-5 and IL-13 in nasal polyp tissue [158,159,160,161].

### 3.2. Helper T Cells (Th Cells)

#### 3.2.1. T Helper 1 (Th1) Cells

Th1 cells are CD4^+^ cells that mainly secrete IL-2, IFN-γ, and TNF-β (tumor necrosis factor β), which are involved in regulating cellular immunity, assisting in the differentiation of cytotoxic T cells, mediating cellular immune responses, and participating in delayed hypersensitivity reactions. IFN-γ, IL-2, and TNF-β can drive inflammatory responses [162].

*S. aureus* infection can induce Th1-type inflammatory responses in different diseases. Enterotoxin B (SEB) induces Th1 activation in chronic sinusitis; α-toxin promotes Th1 activation via CXCL10 in AD [48,49,93,163]. Activated Th1 cells release the cytokine IFN-γ. IFN-γ is a driver of chronic inflammation in the chronic phase of AD, and the overexpression of IFN-γ can lead to recurrent inflammation and pruritus, causing lichenoid degeneration of the skin [13,164]. In contrast, psoriasis is also a chronic disease mediated by Th1 cytokines, and IL-12 mediates the differentiation of Th1 cells [165].

#### 3.2.2. Th2 Cells

Th2 cells (which are CD4+ T cells) are important cells in the type 2 inflammatory pathway. th2 secretes IL-4, IL-5, and IL-13 and stimulates type 2 immunity, as evidenced by the production of high levels of IgE and eosinophils. Type 2 immunity is a specific immune response that includes both innate and adaptive immunity and contributes to the formation of an immune barrier on the mucosal surface to a clear response to pathogens [166]. Th2 cells are formed by the induced differentiation of Th0 cells by IL-4 produced by basophils, eosinophils, mast cells, natural killer cells (NK), or already differentiated Th2 cells.

In AD infection with *S. aureus*, keratinocytes can express TSLP, IL33 [119,154], and IL-19 [167], which can induce the Th2 expression of IL-4, IL-10, IL-13, and IL-31, causing an inflammatory response. IL-4 and IL-31 play important roles in itchiness [158,159,167]. *S. aureus* was found to induce the expression of IL-8, IL-19, and IL-22, which induced increased expression of Th2 cytokines [131,167,168]. *S. aureus* PGN action on LCs induces Th2 cells in the skin via CCL17, and it induces IL-18 stimulation of CD4^+^ T cells to produce IL-4 to induce Th2 cytokine expression [130,131]. *Staphylococcal* α-toxin mediates the expression of Th2 cytokines IL-4 and IL-13 through STAT6, leading to increased keratin-forming cell death [169].

It was found that antigen-specific regulatory T (Treg) cells can also induce Th2 cell proliferation [170]. *S. aureus* PGN action on LCs induces Th2 cells in the skin via CCL17, and it induces the IL-18 stimulation of CD4^+^ T cells to produce IL-4, which induces Th2 cytokine expression [130,131]. In addition, *staphylococcal* α-toxin mediates the expression of Th2 cytokines IL-4 and IL-13 through STAT6, leading to increased keratin-forming cell death [169]. SEB can also act on eosinophils, monocytes, and Treg cells to induce Th2 cytokine expression and induce the Th2 proliferation and secretion of IL-10 to aggravate skin inflammation [170,171]. In asthma, eosinophils induce the activation of Th2 and express IL-5 in conjunction with basophils [172]. In chronic sinusitis, Th2 cytokines can activate B cells, release IgE, and indirectly mediate eosinophil inflammation [7]. Th2 cytokines can reduce filamentous protein expression in keratinocytes, further exacerbating the disruption of skin barrier function [173]. Th2 expresses IL-4, IL-10, IL-13, and IL-31, inducing epidermal thickening, sensitization, inflammation, and pruritus [174,175]. Th2 also reduces the expression of AMP, HBD-2, HBD-3, filamentous polymerase, and epidermal proteins, further promoting the proliferation of *S. aureus* and exacerbating flora imbalance. The activation of keratinocytes increased the expression of endogenous serine proteases, induced inflammatory responses and tissue damage, and released various cytokines. Serine protease V8 and the serine protease strip toxin cleave corneal adhesion proteins, including DSG-1, leading to increased desquamation [158,159].

#### 3.2.3. Th17 Cells

Th17 is a newly discovered subpopulation of T cells that secrete IL-17, which is important in autoimmune diseases and the body’s defense response. Transforming growth factor β (TGF-β), IL-6, IL-23, and IL-21 play active roles in the differentiation and formation of Th17 cells, while IFN-γ, IL-4, cytokine signaling (IFN-γ), IL-4, suppressor of cytokine signaling 3 (Socs3), and IL-2 inhibit its differentiation [176].

Th17 cells have a crucial role in host defense. Dysregulated Th17 responses mediate various autoimmune and inflammatory diseases. IL-6, TGF-β, and IL-23, secreted by macrophages and DCs, induce the activation and differentiation of Th17. Th17 cells are recruited into the skin and interact with keratinocytes and fibroblasts to promote epidermal tissue repair [164]. Th17 cells also express IL-17, IL-22, and IL-26. IL-17 can coordinate local tissue by upregulating pro-inflammatory cytokines and chemokines (including IL-1β, IL-6, TNF-α, granulocyte-macrophage colony-stimulating factor (GM-CSF), KC/CXCL1, MCP-1/CCL2, MIP-2/CXCL2, MCP-3/CCL7, and MIP-3α/CCL20, and matrix metal proteinases (MMPs)), enabling the migration of activated T cells through the ECM [177,178,179].

*S. aureus* triggers a strong Th17-type response in skin effector T cells, inducing Th17 to produce IL-17A, IL-17F, and IL-22 [180]. *S. aureus* promotes keratinocyte expression of IL1β, promotes differentiation of Th17, and promotes skin inflammation [181]. SEB induces IL-21 secretion by Follicular helper T (Tfh), induces differentiation of Th17, and acts in nasal polyp disease [89,90].

In septic arthritis caused by *S. aureus* infection, *S. aureus* activates immune cell-specific macrophages and DCs to release pro-inflammatory mediators, such as TNF-α, IL-1β, IL-6, and IL-21, which induce RORγt, RORγt, and CD4 T cells to differentiate into Th17 cells [182,183,184]. Th17 cells produce the pro-inflammatory cytokine IL-17, which directly stimulates arthritic inflammation by binding to receptors on immune cells, stimulating the production of more pro-inflammatory cytokines, chemokines, and other inflammatory mediators, including NO and MMPs, while TNF-α, IL-1β, and IL-6 upregulate MMPs, which can promote cartilage degradation and enhance joint destruction [185,186].

IL-17 activates macrophages to express TNF-α and IL-1β as well as fibroblasts to produce IL-6, IL-8, and MMPs; it also stimulates blood endothelial cells to produce chelators and the p38 MAPK-dependent expression of VCAM-1 and ICAM-1, which contribute to immune cell evasion from blood to tissue [180]. In mouse studies, ICAM-1 was upregulated on endothelial cells in the diseased skin of CD18 (β2-integrin) (CD18hypo) mice, whereas in vivo *S. aureus* extracellular adhesion protein prevented T cell vascular migration to the inflamed skin of CD18 (β2-integrin) (CD18hypo) mice but did not inhibit their proliferation and activation [109]. IL-17 also promoted epithelial cell stimulation by IL-8 and granulocyte colony-stimulating factor (G-CSF), induced neutrophil migration and activation, and synergistically enhanced the induction of TNF-α. IL-22 not only triggers a pro-inflammatory response, but also inhibits terminal differentiation of keratinocytes and induces Th2-type inflammation [168].

In psoriasis, IL-17 can activate macrophages to express TNF-α and IL-1β, thereby inducing fibroblast activation [180]. In inflammatory areas of the skin, Th17 can increase the inflammatory response of the skin, and IL-17A can strongly induce IL-19 expression in keratinocytes. IL-19 also induces the expression of Th2 cytokines [167,187].

#### 3.2.4. Tregs

Tregs, characterized by an expression of the forkhead transcription factor FOXP3 and IL-2R α chain CD25, play a central role in maintaining tolerance to self and preventing an overexcited inflammatory response to infection [188]. *S. aureus* induces the proliferation of effector memory T (Tem) cells and the activation of Treg cells and Th17 responses through cutaneous LCs [189,190]. Th17 and Treg cells antagonize each other, and Th17/Treg cell imbalance can trigger or exacerbate the disease process in AD [191]. After superantigenic stimulation of SEs, Treg cells lose their immunosuppressive activity, and a Treg induces Th2 proliferation and the secretion of IL-10 to aggravate the skin inflammatory response [170].

#### 3.2.5. Tfh Cells

Tfh cells are a specialized subset of CD4^+^ T cells located in B cell follicles that induce B and T cell interactions and release cytokines to promote germinal center (GC) formation, promote the differentiation of GC B cells into memory B cells or plasma cells, and drive the maturation of high-affinity antibodies [192]. The Tfh cells express CXC chemokine receptor 5 (CXCR5) [193], ICOS [194], programmed death-1 (PD-1) [195], cytotoxic T lymphocyte antigen 4 (CTLA-4) [196], B-cell lymphoma-6 (BCL-6) [197], and IL-21 [90].

In *S. aureus*-infected chronic rhinosinusitis with nasal polyposis (CRSwNP), Tfh can migrate into lymphoid aggregates (lymphoid aggregates resemble germinal centers) and interact with B cells to induce the proliferation and differentiation of naive B cells to plasma cells [89]. Tfh is also involved in the B cell-mediated immune response induced by the *S. aureus* vaccine [198]. In nasal polyposis, IL-21 expression was increased after SEB stimulation. IL-21 can also induce the differentiation of Th17 [89,90].

However, there has been minimal research on this aspect of the mechanism of Tfh in *S. aureus* infection, and future research in this direction may allow the mechanism of *S. aureus* infection to be further deciphered and allow for more comprehensive counseling for clinical treatment.

#### 3.2.6. Th9 Cells

Recently, researchers have identified another novel and unique population of immunomodulatory cells. Th9 cells were initially thought to be a subpopulation of Th2 cells that could produce IL-9 [199]. Th9 cells have been shown to act on many cell types associated with asthma, including T cells, B cells, mast cells, eosinophils, neutrophils, and epithelial cells, and therefore may be important in the pathophysiology of allergic asthma [200]. Moreover, Th9 has been found to be a major factor in regulating immunity to autoimmune diseases [201] and tumor immunity [202].

There are few studies on this aspect of the mechanism of *S. aureus* infection promoting Th9-type inflammatory response, which may be a direction for further research.

### 3.3. ILCs

ILCs are lymphocytes that do not express the diverse antigen receptor types expressed on T and B cells [203]. ILCs comprise a heterogeneous population of immune cells that maintain barrier function and can initiate a protective or pathological immune response in response to infection [204].

ILCs can be divided into ILC1, ILC2, and ILC3 types. In cell-mediated effector immunity, ILC1, ILC2, and ILC3 are involved in type 1, type 2, and type 17 immune responses, respectively [205]. *S. aureus* and *S. aureus* muramyl dipeptide (MDP) can activate ILC2 and promote type 2 immune response [161]. In AD, PSMα induces the production of ILC3, which is involved in the mediation of skin inflammation [5]. In patients with AD infection with *S. aureus*, the expansion of ILC3 is involved in mediating skin inflammation [13].

However, little is known about the relationship between *S. aureus* infection and ILCs. This could be another direction in the study of the mechanism of *S. aureus* infection in the future.

### 3.4. Macrophages

Macrophages are cellular components of the innate immune system and are present in almost all tissues, contributing to immunity, repair, and homeostasis [206]. When *S. aureus* infects humans, epithelial cells recognize the invading *S. aureus* through pathogen recognition receptors (PRRs), which induce the production of pro-inflammatory cytokines and chemokines, leading to the recruitment and activation of phagocytes, including GM-CSF, G-CSF, IL-1β, IL-6, and IL-8 [8].

Phagocytosis of *S. aureus* triggers the TLR2-dependent signaling and activation of the NLRP3 inflammasome, leading to the recruitment of ASC and the activation of cystathione-1, causing cells to release cytokines IL-1β and IL-18 and inducing apoptosis [207,208]. *S. aureus* was found to survive and replicate in macrophages, which deliver nutrients to lysosomal-engulfing *S. aureus* to promote bacterial growth, and promote bacterial persistence during infection by limiting reactive oxygen species (ROS) and RNS production by macrophages through lipoic acid synthesis [209,210].

### 3.5. DCs

DCs are the most powerful specialized antigen-presenting cells in the body and are highly efficient in the uptake, processing, and presentation of antigens. Immature DCs have a strong migratory capacity and mature DCs can effectively activate initial T cells and are central to the initiation, regulation, and maintenance of the immune response. They are usually found in small numbers in contact with external skin and their immature forms can be found in the blood. When activated, they move to the lymphoid tissue to interact with T and B cells to stimulate and control the appropriate immune response.

*S. aureus* activates macrophages and DCs to release pro-inflammatory mediators, such as TNF-α, IL-1β, IL-6, and IL-21, which induce RORγt to produce IL-17A, IL-17F, and IL-21 and induce CD4 T cells to differentiate into Th17 cells [182,183,184].

TSLP released by keratinocytes in AD is a potent activator of DCs, triggering the production of Th2-attracting chemokines, such as CCL17/TARC and CCL22/MDC, and inducing Th2 differentiation through upregulation of these cells by OX40L, which activates Th2 cells [9]. In experiments with psoriatic mice, *S. aureus* Eap was found to disrupt cell–cell contacts between T cells and DCs in vitro, blocking T cell extravasation into inflamed skin to inhibit psoriasis [109].

LCs are the major DCs in the normal epidermis, and DCs are the major antigen-presenting cells [211]. Epidermal exposure to *S. aureus* induces the proliferation of effector memory T (Tem) cells and limited Treg cell activation and Th17 responses, and LCs directly interact with *S. aureus* via the pattern recognition receptor langerin (CD207), which interacts directly with *S. aureus* [189,190]. LCs express TLRs that recognize bacterial and viral products, and the TLR2-mediated transduction of *S. aureus*-derived signals is severely impaired in LCs with AD skin [212]. The DC-mediated blockade of human T cell activation and proliferation, PVL, targets DCs to blunt CD4^+^ T lymphocyte activation and kills DCs, leading to impaired T cell responses and increased infection [213].

### 3.6. Mast Cells

Mast cells are granulocytes that disintegrate to release granules and the substances in the granules. Such granules in the blood contain heparin, histamine, and 5-hydroxytryptamine, which can drive tachyphylactic allergic reactions (inflammation) in tissues, especially in asthma. Mast cells are the main effector cells of inflammation, and mast cells act as antigen-presenting cells and induce Th1 and Th2 cell development.

In patients with AD, *S. aureus* infection leads to an increase in the number of mast cells, which proliferate within mast cells and mediate Th1 cell development as well as the development of chronic inflammation, leading to the release of Th1 cytokines and the upregulation of IFN, manifested as edema within the lamina propria [214,215]. In mast cells, PGN from *S. aureus* stimulates mast cells in a TLR2-dependent manner, producing TNF-α, IL-4, IL-5, IL-6, and IL-13, and *S. aureus* also triggers TNFα and IL-8 release by binding to CD48 [132,133].

The δ toxin induces mast cell degranulation, which is dependent on intracellular PI3K activation and free calcium ion influx. δ toxin-induced mast cell degranulation differs from conventional IgE cross-linking in that this action does not require the presence of an antigen. IgE enhances δ toxin-induced mast cell degranulation, promotes IgE and IL-4 production, and leads to skin inflammation. However, AD is a chronic inflammatory skin disease caused by mast cells, leading to immunoglobulin-E (lgE)-mediated hypersensitivity [74,147].

*S. aureus*-expressed SAgs can also trigger mast cell degranulation via IgE and FcεR, and SEs may also trigger direct histamine release from mast cell degranulation via unknown receptors [10]. In asthma, the airways are hyperreactive, and mast cells release histamine and various cytokines that attract the accumulation of eosinophils, causing epithelial cell damage and respiratory distress [216,217].

Although mast cells may help clear the infection, *S. aureus* may use mast cells to evade detection and immune clearance.

### 3.7. Neutrophils

Neutrophils are a type of myeloid leukocyte and some of the major responders in acute inflammation. Activated neutrophils mediate inflammation by synthesizing and secreting cytokines, chemokines, leukotrienes, and prostaglandins. Neutrophils synthesize and secrete the chemokine CXCL8 to recruit more neutrophils and express IL-1, IL-6, IL-12, TGF-β, and TNF-α, reactivating neutrophils and other cells of the immune system [218].

*S. aureus* can recruit and activate neutrophils at the site of infection [219]. In AD and pneumonia, PSMα can induce the expression of IL-1α and IL-36 and induce neutrophil death, leading to disease exacerbation [67,68]. γδ T cells mediate IL-17 responses and induce neutrophil recruitment, pro-inflammatory cytokines IL-1α, IL-1β, and TNF, and host defense peptides, while rapid neutrophil recruitment enhances *S. aureus* colonization in the skin [11]. IL-17 also promotes IL-8 and G-CSF stimulation of epithelial cells, induces neutrophil migration and activation, synergistically enhances TNF-α induction, and increases antimicrobial peptide (AMP) production [180].

Neutrophils induce IL-20 expression, which contributes to psoriasis, wound healing, and anti-inflammatory effects [165]. *S. aureus* LTA promotes the expression of neutrophil factors TNF-α and IL-8. IL-8 also attracts neutrophils to accumulate in the intestine, leading to the activation and attraction of polymorphonuclear leukocytes (PMN), causing an inflammatory response in the intestine during food poisoning [23]. The α toxin produces IL-1β via TLR2, NOD2, FPR1, and ASC/NLRP3 inflammasome induced by neutrophils, and IL-1β can induce thymic stromal lymphopoietin and contribute to abscess formation [178,220]. *S. aureus* α toxin produces IL-1β via TLR2, NOD2, FPR1, and ASC/NLRP3 inflammasomes induced by neutrophils expressing IL-1β [220]. In GPA, the activation of neutrophils, by expressing neutrophil extracellular trap products (NET-derived products), plays a role in the disease [221].

### 3.8. Eosinophils (Eos)

Eosinophils are considered to be the effector cells associated with infection and the cause of tissue damage. Eosinophils can express a range of ligand receptors that play a role in cell growth, adhesion, chemotaxis, degranulation, and intercellular interactions. Eosinophils synthesize, store, and secrete cytokines, chemokines, and growth factors. Eosinophils can function as antigen-presenting cells and can regulate the immune system [222].

In AD, *S. aureus* produces exotoxins that can amplify the allergic response by directly activating other immune cells, such as eosinophils. Eosinophil inflammation can be induced by SEB. Eosinophils lead to epidermal damage, tissue swelling, and inflammatory cell recruitment through the release of toxic mediators, including eosinophil cationic protein (ECP), major basic proteins, eosinophil-derived neurotoxins, and eosinophil peroxidase (EPO) [223,224]. In addition, activated eosinophils can promote antimicrobial defense by releasing mitochondrial DNA associated with granule proteins [12].

In asthma, eosinophils are not only involved in the release of granulins, lipid mediators, ROS, cytokines, and growth factors that trigger the Th2 response [172]. The IL-5 cytokines of Th2, GM-CSF, and IL-3 induce the eosinophil response and eosinophil maturation. IL-3 and GM-CSF also induce eosinophil recruitment [225,226].

In CRSwNP, *S. aureus* can also cause eosinophil inflammation by inducing IgE [7].

### 3.9. Basophils

Basophils are a type of leukocyte that originate from bone marrow hematopoietic pluripotent stem cells that differentiate and mature in the bone marrow and enter the bloodstream. Basophils are important cells for *S. aureus* respecting the virulence of AD and asthma [3,216].

*S. aureus* SAgs, including SEA, SEB, SEC, and TSST-1, also directly activate B lymphocytes and induce specific IgE-dependent mast cells and basophil degranulation to release histamine, further exacerbating AD and promoting adaptive cellular and humoral type 2 immunity [13].

NOD2 and TLR2 ligands trigger basophil activation by interacting with dermal fibroblasts in AD-like skin inflammation [227]. *S. aureus* was found to induce skin basophil aggregation and increase IL-4 expression. Basophil-derived IL-4 inhibited skin IL-17A production by TCRγδ+ cells and promoted *S. aureus* infection of the skin. Basophils secrete IL-6 to promote Th17 responses and inhibit the IL-17A production of IL-4 via STAT6 inhibition of the IL-17A promoter to promote *S. aureus* infection [228], while Toll-like expressing receptor-expressing epidermal keratinocytes recognize invasion and respond to LTA by inducing the expression of cytokines such as TSLP, which leads to basophil recruitment and IL-4 production [126].

### 3.10. B Cells

In inflammatory and neoplastic diseases, B cells play a regulatory role by secreting regulatory cytokines, such as IL-10, or by relying on the secretion of antibodies. B cells can act in concert with other immunomodulatory cells, such as Treg cells. TSST-1 can induce B cell apoptosis [229]. LTA inhibits LPS-induced B cell proliferation by reducing ERK phosphorylation through the TLR2 signaling pathway [230]. The *S. aureus*-induced activation of Th2 triggers B cells to produce IgE in response to allergens and autoantigens, indirectly mediating eosinophil inflammation [7]. Tfh and ILCs can also induce B cell differentiation [89,216].

In the disease response to *S. aureus* infection, there are many other cells involved besides those mentioned above. As a major human pathogen, *S. aureus* induces cell activation in various cell types, releases various cytokines, and causes apoptosis, which is important in *S. aureus* infections. However, the cellular mechanisms are not yet clear, including the role of Th9, Tfh, and ILCs in the pathogenesis of *S. aureus*. This may be another direction for studies on *S. aureus* infection in the future.

A summary of the inflammatory cells activated by *S. aureus* in several diseases is shown in Figure 1.

## 4. Types of Host Cell Death Caused by *S. aureus*

*S. aureus* can induce cell death modalities, such as pyroptosis, apoptosis, necroptosis, and autophagy, which in many ways have substantial impacts on the pathogenesis and clinical manifestations of *staphylococcal* disease. An increasing number of studies have shown that the various cell death modalities are interrelated [231]. This suggests that the various modes of cell death do not exist in isolation.

### 4.1. Pyroptosis

#### 4.1.1. Activation of Inflammasomes Caused by *S. aureus*

The inflammasome is a caspase-1-activated multiprotein platform that responds to many exogenous and endogenous danger signals [232]. Inflammasomes may be important in the development of diseases and may be an important trigger for disease development. Inflammasomes activate the caspase protease caspase-1, which subsequently activates the proinflammatory cytokines IL-1β and IL-18 and induces pyroptosis [233,234].

Inflammasomes can recognize the uncontrolled release of damage-associated molecular patterns (DAMPs) or PAMPs through pattern recognition receptors (PRRs) [235]. Their main components include NLRP1 [236], NLRP2 [237], NLRP3 [238], NLRP4 [239], NLRP6 [240], NLRP7 [241], NLRP12 [242], NLRC4 [243], AIM2 [244], IFI16 [245], and Pyrin [246,247]. Among these, NLRP3 [35] and NLRP6 [125] play an important role in *S. aureus* infection.

*S. aureus* acts through two signaling pathways to activate inflammasomes to promote AD: (1) Signal 1: *S. aureus* acts on TLRs and TNFR1, interacts with NOD2, activates NF-kB, induces gene transcription, produces Pro-IL-1β, and synthesizes raw materials for the inflammasomes. (2) Signal 2: PFTs, purine receptors, and lysosome rupture act together to cause potassium ion efflux, which, together with the inflammasome raw materials produced by signal 1, forms the inflammasome. Inflammasomes activate caspase-1, which subsequently activates the proinflammatory cytokines IL-1β and IL-18 and induces pyroptosis [233,234]. The NLRP3 inflammasome was found to promote IL-1β activation at the site of *S. aureus* skin infection in mice by mediating neutrophil recruitment to the site of skin infection [220,248].

Inflammasomes can suppress the expression of TSLP, which regulates Th1 and Th2 cells. The gene expression of NLRP3 and ASC is significantly reduced during Th2 cytokine (IL-4, IL-5, and IL-13) stimulation in human keratinocytes. Therefore, NLRP3 inflammasome activity can be inhibited by suppressing the expression of NLRP3, ASC, inflammatory cystein-1, and IL-1β in keratinocytes [249,250].

Munoz-Planillo et al. [35] found that SEs activated the NLRP3 inflammasome in macrophages. When bound to bacterial lipoproteins, enterotoxins SEA and SEB activated the inflammasome. This activation was not dependent on the P2X7 receptor or the TLR junction MyD88, suggesting that SEs may bypass the P2X7 receptor activation pathway for the NLRP3 inflammasome.

#### 4.1.2. Pyroptosis

Pyroptosis is a new form of programmed cell death caused by the inflammasome, which is characterized by the dependence on inflammatory cysteases (mainly caspase-1, 4, 5, and 11) and accompanied by the release of large amounts of pro-inflammatory factors. The morphological features, occurrence, and regulatory mechanisms of pyroptosis death are different from other cell death modalities, such as apoptosis and necrosis. The activation of caspases by the inflammasome leads to the cleavage of gasdermin proteins, the activation of gasdermin proteins, the translocation of activated gasdermin proteins to the membrane, the formation of pores, cell swelling, cytoplasmic outflow, and finally cell membrane rupture and cell death. There are also classical and non-classical pathways for cell death. The classical pathway, the caspase-1 pathway, recruits and activates caspase-1 through inflammasomes that sense danger. Caspase-1 cleaves and activates inflammatory factors, such as IL-18 and IL-1β, which cleave the N-terminal sequence of GSDMD and bind to the membrane to create membrane pores, leading to pyroptosis. In the non-classical pathway, human-derived caspase-4,5 and murine-derived caspase-11, on the other hand, can be activated by direct contact with bacterial LPS, etc., which then cleaves the GSDMD and indirectly activates caspase-1, triggering pyroptosis. The GSDMD of the Gasdermin protein family is cleaved by inflammatory cystathionases and exhibits pore-forming activity to drive pyroptosis [251]. *S. aureus* can activate NLRP3 and lead to MAC-T cell pyroptosis via the K^+^ efflux pathway, and the homeostatic phenotype of resident macrophages can also orchestrate the early inflammatory response via the inflammasome pathway and subsequent pyroptosis [252,253]. It was found that the *S. aureus* associated-PAMPs, i.e., LTA [125], triacyl lipopetides [124], diacyl lipoproteins [151], and PGN [140], and *S. aureus* toxins, enzymes, and effectors, i.e., PVL [62],α-Hla [143], Hlb [34], and Hlg [35], can mediate pyroptosis [16].

*S. aureus*, which is phagocytosed by macrophages, increase the expression of NLRP3 and caspase-1 (Casp-1 p20), the accumulation of GSDMD-NT, and the release of IL-1β and IL-18 in macrophages and induce macrophage pyroptosis [254]. The LTA-induced activation of the NLRP6 inflammasome promotes the expression of caspase-1, IL-1β, and IL-18 in macrophages, inducing host cell pyroptosis [125]. In sepsis and venous infections, the shedding of lipopeptides from bacterial surfactants induces the activation of TLR 2 [69]. Lipoproteins from *S. aureus* are essential for growth and immunity. The lysozyme promotes activation of the NLRP3 inflammasome in the presence of lipoproteins and the induction of caspase-1 [35,123]. Furthermore, PGN strongly induces NLRP3 inflammasome expression in macrophages in *S. aureus* skin infections [140]. PVL exposure activates NLRP3, and binding to monocytes and macrophages leads to the release of the caspase-1-dependent pro-inflammatory cytokines IL-1β and IL-18, which induce macrophage death and exacerbate pneumonia [58,62]. It was found that *S. aureus* also induced keratinocyte pyroptosis via a caspase-1-dependent form of inflammatory cell death [143]. Activation of the NLRP3 inflammasome and caspase-1 by *S. aureus* toxin Hla induces the production of IL-1β and IL-18 and pyroptosis [14].

### 4.2. Necroptosis

Necroptosis, a form of programmed inflammatory cell death, was originally identified as an alternative to apoptosis following the involvement of death domain receptors. The classical necrosis pathway consists of RIPK1–RIPK3–MLKL. The necrosis pathway is triggered downstream of the death structural domain receptors (e.g., TNF receptors and Fas) and TLR3/4. Active RIPK1 is recruited in oligomeric complexes containing FADD, caspase-8, and caspase-10. In the absence of caspase-8, RIPK1 recruits and phosphorylates RIPK3 to form a RIPoptosome, which then phosphorylates MLKL to form a necrosome. MLKL oligomers translocate to phosphatidylinositol phosphate (PIP)-rich patches at the plasma membrane, forming macropores that lead to necrotic cell death by allowing ion inward flow, cell swelling, and membrane lysis, followed by the uncontrolled release of intracellular material [255].

PVL [63], Hla [43], and PSMs [69] induce the necroptosis of host cells after *S. aureus* infection [16]. In pneumonia and psoriasis, *S. aureus* infection induces necroptosis through RIP1/RIP3/MLKL signaling. Phagocytosis by *S. aureus* triggers the programmed necrosis of neutrophils. Bacterial pathogens increase their chances of survival by disrupting host programmed cell death (PCD). *S. aureus* is recognized by homologous host receptors, and the bacteria adhere and enter non-phagocytic cells, triggering programmed cellular necrosis through either intrinsic pathway mitochondrial damage or the extrinsic pathway in response to cell surface receptor engagement [16,63,255,256]. TNF and TLR3/TLR4 activation triggered necroptosis through a downstream signaling complex containing RIPK1, RIPK3, and MLKL. Whereas c-Jun N-terminal kinase (JNK1 and JNK2) activity significantly enhanced programmed cell death induced by TNF and TLRs, RIPK1 and RIPK3 promoted cell death-independent JNK activation in macrophages [257]. However, it has also been suggested that programmed cellular necrosis is a host cell death mechanism independent of the caspase and that glycolysis and mitochondrial ROS production are sufficient to induce programmed cellular necrosis [258]. PSMs can also induce cell death through formylpeptide receptor 2 (FPR2) by inducing the secretion of TNF-α [68].

### 4.3. Apoptosis

Apoptosis is a programmed cell death process that relies on a cascade of caspase endopeptidase activity called cystathionase. As a major human pathogen, *S. aureus* can induce apoptosis during infection through various pathways, which are also important in *S. aureus* infections [259]. It has been shown that apoptosis is essential in certain diseases caused by *S. aureus*, such as AD and sepsis [260]. Apoptosis of the host immune system may promote *S. aureus* infection, and the apoptosis of tissue cells may also reduce the immune response by affecting cytokine production and T-cell differentiation [253]. Thus, apoptosis can significantly influence the pathogenesis of *S. aureus*. With the exception of certain cellular components of *S. aureus* that may promote apoptosis (e.g., SpA [115]), most apoptotic processes caused by *S. aureus* are derived from different species of secreted toxins. Among the known *S. aureus* bacteria that have many virulence components that activate FADD, MyD88, RIPK3, caspase-8, and caspase-3 showed pro-apoptotic activity. The components include α-toxin [47], PVL [64], SEs (SEA [148], SEB [15], SEM [98], and SEH [96]), SpA [115], EsxA [101], Coa [104], Nuc, AdsA [108], SspB [85], Hla [144], and PGN [261].

*S. aureus* plays a role in osteomyelitis nephrapostasis, cancer, and bacteremia through apoptosis [16]. In pneumonia, *S. aureus* α-toxin inhibits the ability of alveolar macrophages (AMs) to clear neutrophils at the site of infection, induces the activation of caspase-3 in lymphocytes and monocytes, mediates cell death, activates caspase-8 and -9, and induces the release of cytochrome C [144,262].

PVL is an important virulence factor for the induction of apoptosis, acting on NLRP3 to induce macrophage death and IL-1β secretion during the course of pneumonia disease; PVL induces the release of the apoptotic proteins cytochrome c and Smac/DIABLO in mitochondria [62,65].

SEA triggers TCR-mediated cell death in CD4^+^ T cells [148]. SEB induces the differential expression of Rnd3 and RhoA in proximal renal tubular epithelial cells and induces actin stress fiber assembly and apoptosis [15]. In bovine mammary epithelial cells infected with *S. aureus*, SEM induces apoptosis by decreasing the mitochondrial membrane potential (ΔΨm) as well as the intracellular ATP concentration and activating caspase-3 [98]. SEH was also found to induce apoptosis in bovine mammary epithelial cells in vitro [96].

The SpA of *S. aureus* induces the release of apoptotic TNF-α and NO, induces apoptosis in EAC cells by upregulating pro-apoptotic factors (p53 and Bax) and downregulating anti-apoptotic factors (Bcl-2), induces the activation of caspase-3, and induces apoptosis in tumor cells, thus playing a role in cancer [115]. Bcl-2 inhibits neutral sphingomyelinase and ceramide-induced cytochrome c release, resulting in a delay in apoptosis; however, Bax accelerates cytochrome c release and caspase activation to induce apoptosis [116]. SpA is also closely associated with the disease process of osteomyelitis due to *S. aureus* infection [117].

In the immune response to *S. aureus* infection, EsxA regulates the production of cytokines and apoptotic processes, induces apoptosis, and, together with EsxB, mediates the release of *S. aureus* from host cells [101,102].

In bacteraemia and renal abscesses, *S. aureus* can induce caspase-3 production by secreting Nuc and AdsA. Neutrophil extracellular traps produce cytotoxic deoxyadenylation (dAdo) to induce caspase-3 production. Caspase-3 can trigger the apoptosis of macrophages [107,108].

SspB selectively cleaves CD11b on phagocytes, induces the phagocyte expression of phosphatidylserine and membrane-linked protein I, and induces apoptosis [85].

PGN activates the TLR2-mediated PP2A-ASK1-JNK-AP-1-C/EBPβ cascade, the macrophage expression of COX-2, and IL-6 expression [134,135]. PGN can also induce IL-1β, IL-8, and TNF-α expression through TLR2 signaling and induce the apoptosis of THP-1 cells [136].

### 4.4. Autophagy

Autophagy is the process of isolating cellular components into lysosomes for degradation and is an evolutionarily conserved mechanism for adapting to unfavorable microenvironmental conditions [263]. In *S. aureus* infection, host autophagy occurs for *S. aureus* defense. The autophagy receptor induces selective autophagic degradation of *S. aureus* by specifically recognizing the entering cells. *S. aureus* is recruited to the autophagosome membrane by early phagosomes bearing Rab5 and Rab22b to LC3 labeled by Rab7 and LAMP-1 in the presence of Hla. *S. aureus* can inhibit the fusion of autophagosomes with lysosomes to form autolysosomes and thus replicate within the autophagosome without being killed [264,265]. However, *S. aureus* can inhibit autophagosome maturation and fusion with lysosomes, escape from autophagosomes into the cytoplasm, and lead to host cell death, which does not rely on caspase [266]. In contrast, the pore-forming toxin Hla can promote autophagy by decreasing intracellular cAMP levels, leading to decreased activation of the protein Epac [145,146]. *S. aureus* induces inflammatory response and autophagy in macrophages by TLR2 through multiple signaling pathways, including the JNK and PI3K signaling pathways [267,268]. It was found that unmethylated cytosine-phosphate-guanine (CpG) in the JNK/P38 signaling pathway promotes phagocytosis and autophagy in *S. aureus*-stimulated macrophages [139]. Endothelial cells with defective autophagy have been found to be more susceptible to the *S. aureus* α-toxin [36]. In bovine mammary epithelial cells (BMECs), *S. aureus* evades autophagic degradation by inhibiting autophagic flux and disrupting lysosomal function [269]. In PMN, intracellular survival of *S. aureus* was associated with the accumulation of autophagic flux markers LC3-II and p62 [36].

In sepsis and pneumonia, the PGN of *S. aureus* and TLR2 ligands also acts on the central nervous system, activating microglia and inducing autophagy and autophagy-dependent cell death [17]. In contrast, *S. aureus* strains with high levels of Agr activity are able to block the autophagic flux, leading to the accumulation of autophagosomes and promoting the spread of *S. aureus* infection [270]. MRSA can promote the degradation of inflammasome components through autophagy and promote their persistence in keratinocytes [271].

### 4.5. Ferroptosis and Cuprotosis

Ferroptosis is a novel form of programmed cell death that is iron ion-dependent and distinct from apoptosis, necroptosis, and autophagy. Ferroptosis is an iron-dependent form of regulated cell death caused by unrestricted lipid peroxidation and subsequent membrane damage. Ferroptosis is induced through an extrinsic pathway regulating transporter protein initiation and an intrinsic pathway blocking the expression or activity of intracellular antioxidant enzymes [272]. Altering glutathione levels by disrupting caspase countertransport protein (GPX4) or using iron inhibitor-1 specific inhibition can increase intracellular ROS levels in response to free intracellular iron, leading to the oxidative degradation of lipids in cell membranes and oxidative cell death [16].

*S. aureus* can be deregulated by the expression of ROS-dependent, and loss of function mutations within the phagocytic NADPH oxidase NOX2 can lead to impaired production of ROS [273,274,275]. However, there are few studies on the mechanism of *S. aureus* in iron necrosis, so the virulence mechanism of *S. aureus* could be a direction for future studies.

New research has identified a new mechanism of cell death called cuprotosis, in which excess copper causes mitochondrial proteins to accumulate and drives cell death [276]. Cuprotosis occurs through the direct binding of copper to the lipid acylated component of the tricarboxylic acid (TCA) cycle, leading to lipid acylated protein aggregation and the subsequent loss of iron-sulfur cluster proteins, resulting in proteotoxic stress and ultimately cell death [277]. In contrast, in *S. aureus*, ClpC ATPase regulates the TCA cycle and then regenerates and enters the bacterial death phase after fixation through functional TCA cycle metabolism [278,279]. Whether there is a corresponding link between *S. aureus* and cuprotosis remains to be determined.

A summary of the cell death mechanisms activated by *S. aureus* in several diseases is shown in Figure 2.

## 5. Basic Types of Diseases Caused by *S. aureus* and Virulence Mechanisms

### 5.1. Skin Disease

#### 5.1.1. AD

AD is the most common chronic inflammatory skin disease with a lifetime prevalence of up to 20% and a significant impact on quality of life. AD is characterized by intense pruritus, impaired epidermal barrier function, recurrent eczema, and fluctuations in the course of the disease. The pathogenesis of AD is mainly associated with dysbiosis of the skin microbiota (especially overgrowth of *S. aureus*), systemic immune response, and neuroinflammation associated with itchiness [280,281].

The role of *S. aureus* in AD is unquestionable, and studies have shown that nearly 100% of patients with AD present colonization of the skin with *S. aureus* [282].

In AD, the colonization of the skin by *S. aureus* and its secretion of SAgs/toxins have different virulence mechanisms. In AD, *S. aureus* activates TLR 2 on genes controlled by nuclear factor kappa B (NF-κB), which induces cytokine release from keratinocytes [18]. *S. aureus* has been isolated from AD patients and found to adhere to keratinocytes in a clumping factor B-dependent manner. IL-1β induces the activation of Th17 and the release of IL-17 and induces neutrophil recruitment, S100A8/A9, and chemokine activation [10,70,153]. Moreover, *S. aureus* colonization of the skin mucosa was found to directly induce epithelial cell-derived cytokine release by binding to TLR 2.

Skin *S. aureus* colonization resulting in pH changes may serve as a predictor of the increased severity of AD. Skin pH is tightly controlled by intrinsic factors that limit the abundance of *S. aureus* due to high *S. aureus* load, especially during the growth of *S. aureus* outbreaks [283].

The *S. aureus* α toxin, TSST-1, PSMs, SpA, second immunoglobulin, SEs, and ETs play a role in AD [13]. The *S. aureus* α-toxin induces Th1 responses in AD and promotes Th1 polarization by inducing chemokine (C-X-C motif) ligand 10 (CXCL10) in macrophages [48,49]. It was found that the α-toxin and PVL can also induce apoptosis [229]. TSST-1 binds to CD40 on keratinocytes and upregulates chemokine gene expression, releasing IL-9 and MIP-3α, which activates T lymphocytes and macrophages, leading to immune dysregulation [50]. Staphylococcal epidermal ETs may also contribute to the deterioration of AD [31].

*S. aureus* expresses PSMα, which enhances the survival and dissemination of *S. aureus* in invasive infections and is required to trigger skin inflammation [5,67]. PSMα had a cytolytic effect on neutrophils and enhanced the expression of inflammatory molecules in a mouse model of skin necrosis [67]. PSMα induces the release of IL-36α, and released IL-36α and IL-4 act together to trigger IgE class conversion in B cells and to promote plasma cell differentiation and elevated serum IgE levels [5,70].

*S. aureus* Sbi is the major staphylococcal-derived virulence factor that directly drives IL-33 release from human keratinocytes and promotes AD [119].

*S. aureus* superantigen SEs can directly stimulate skin keratinocytes, LCs, skin vascular endothelial cells, and macrophages, including the expression of pro-inflammatory factors TNF-α, IL-10, intercellular adhesion molecules (I-CAM), and vascular cell adhesion molecules (V-CAM), which promote the entry of inflammatory cells into skin tissue, leading to the development of AD. SEs can also lead to the upregulation of the T lymphocytes of associated antigens, prompting memory T lymphocytes to travel to AD lesions and become activated. The upregulation and activation of T lymphocytes induces and exacerbates the inflammation in the lesions [284]. SEA induced an inflammatory response in HaCaT keratinocytes and HUVECs, inducing a significant upregulation of E-selectin, ICAM-1, MCP-1, IL-6, and IL-8 expression. SEB induced a strong inflammatory response in the skin, inducing the upregulation of ICAM-1 and VCAM-1 in dermal vessels [149]. SEB-mediated skin inflammation is T cell-dependent and promotes IL-17 secretion [180,285].

*S. aureus* expresses the δ-toxin, which stimulates cytokine release from keratinocytes and triggers an inflammatory response in keratinocytes as well as B-cell proliferation and cytokine release [10]. In contrast, Th22 (CD4^+^IL-22^+^IL-17AIFN-γ) cells from AD patients inhibit *staphylococcal* SEA and SEB responses [92]. Th22 expressed increased IL-22, as well as IL-4, IL-5, IL-13, and IL-31.

#### 5.1.2. Psoriasis

Psoriasis is an immune-mediated skin disease stimulated by environmental factors and controlled by multiple genes, usually presenting as scaly erythematous plaques or patches confined to one location or widely distributed throughout the body. Most patients have exacerbations or relapses in the winter and remission in the summer. Studies have found that psoriasis is often accompanied by the colonization of the skin with *S. aureus*, leading to exacerbation of the lesions [286].

IL-17 is the major effector cytokine in the pathogenesis of psoriatic disease [19]. The microbial community on psoriatic skin is very different from that on healthy skin. The loss of community stability and the reduction in immunomodulatory bacteria (e.g., *Staphylococcus epidermidis*) may lead to a higher colonization of pathogens such as *S. aureus*, which may exacerbate skin inflammation along the Th17 axis [168].

Psoriasis is also a Th1-induced chronic disease in which PGN from *S. aureus* induces the expression of human cathepsin LL37 and vascular endothelial growth factor (VEGF) in keratinocytes, activates Th1 cells, and expresses IL-13 [137,138]. *S. aureus* also acts on keratinocytes to induce IFN-γ expression by Th1 and IL-12-mediated differentiation of Th1 [165,287]. *S. aureus* can also induce monocytes to express IL-12, an inducer of the differentiation of Th cells to Th1 [165,288]. In psoriasis, *S. aureus* infection can also be mediated by the RIPK1/RIPK3/MLKL-mediated necrotizing apoptosis of keratinocytes, and RIPK1 mediates keratinocyte death via TNF [256,289].

### 5.2. Respiratory Diseases

#### 5.2.1. Asthma

It is well known that allergic asthma is usually mediated by Th2 cells, leading to the production of cytokines IL-4, IL-5, IL-9, and IL-13. The activation of B cells triggers the release of allergen-specific IgE antibodies and induces the degranulation of eosinophils, mast cells, and basophils, leading to airway smooth muscle contraction, epithelial cell dysfunction, and excessive mucus secretion [216].

SEs can penetrate the mucosal barrier under the right conditions and induce continuous eosinophilic inflammation and IgE formation. In polyp samples containing SEA-specific IgE antibodies, severe eosinophilic inflammation, manifested by the upregulation of IL-5, eosinophil chemokines, eosinophil cationic protein (ECP), and caspase-leukotriene synthesis, was observed, thus contributing to the severity of asthma [91]. The study finds that rapid αβ T-cell responses can coordinate pulmonary innate immunity to SEA [21]. Nasal SEB increases the serum concentrations of IL-4, IL-5, and IFN-γ, while bronchial SEB increases the serum titers of specific IgE and total IgE. Exacerbation of asthma is associated with the elevated expression of IL-5, IL-4, IFN-γ, IL-12 p40, eosinophil chemokine-1, and TGF-β mRNA in the bronchial tubes [93].

*S. aureus* also manipulates airway mucosal immunology at different levels through its proteins, inducing the activation of airway epithelium and the release of TSLP, IL-25, and IL-33, leading to sustained immune responses in DCs and ILC2s and the activation of type 2 immune responses [290,291]. In other studies, it was found that airway epithelial cells can release IL-33 and activate ILCs by their receptor ST2, followed by releasing type 2 cytokines, mast cell degranulation, and the massive activation of local B cells and IgE formation, attracting an accumulation of eosinophils.

Extracellular traps are then released, increasing epithelial cell damage and leading to disease-persistence Charcot–Leyden crystals [216]. Staphylococcal Spls cause an IgE antibody response and stimulate Th2 cytokine production by T cells, whereas *S. aureus* Hla causes low or absent concentrations of reactive Th1/Th17 cytokines [78].

However, the mechanism of cell death in asthma by *S. aureus* infection is still minimally investigated today, so the mechanism of asthma with *S. aureus* infection is worth investigating.

#### 5.2.2. Chronic Rhinosinusitis

Chronic sinusitis is a chronic inflammatory disease of the sinus mucosa, and *S. aureus* is found to colonize the mucosa in most patients with chronic sinusitis with nasal polyps. *S. aureus* binding to TLR 2 directly induces the release of epithelial cell-derived cytokines and promotes the expression of TSLP, IL-5, IL-13, and IL-33 in nasal polyp tissue [160]. TSLP may also act directly on ion channel TRPA1-positive sensory neurons to trigger powerful pruritic behavior [157]. SEB induces Th1 and Th2 pro-inflammatory responses in nasal polyps and increases the serum concentrations of IL-4, IL-5, and IFN-γ [93,163]. The α-toxin induces nasal polyp cells to produce large amounts of IL-5, IL-13, IFN-γ, IL-17A, and IL-10, which are involved in the inflammatory response [51].

In patients with chronic CRSwNP, superantigenic SEs activate T cells to release the Th2 cytokines IL-4, IL-5, and IL-13, which activate antigen-specific B cells and produce polyclonal IgE. IgE acts on mast cells to release chemokines (eosinophil chemokines), leading to eosinophil inflammation [7]. In contrast, T lymphocytes located in nasal polyp tissue have higher T cell receptor V-β amplification, enhancing polyclonal IgE production and contributing to sustained Th2 inflammation [292].

In chronic sinusitis, insufficient IFN-γ-induced autophagy can lead to the p62-dependent apoptosis of epithelial cells, while TNF-α and IFN-γ-induced necroptosis may promote the production and release of numerous pro-inflammatory cytokines and lead to neutrophil infiltration to exacerbate inflammation in CRSwNP [293,294]. SEB can also stimulate IL-21 expression, and IL-21 can also induce Th17 differentiation [89,90].

#### 5.2.3. Pneumonia

In pneumonia with *S. aureus* infection, staphylococcal Hla is an important virulence factor in severe *S. aureus* pneumonia [22]. *S. aureus* Hla evades ATP and P2X7 receptors in the presence of lipoproteins to induce cystathione-1 activation via NLRP3 [35]. The activation of inflammasomes induces the secretion of pro-inflammatory cytokines IL-1β and IL-18, which induce cell death and lead to the release of endogenous pro-inflammatory molecules, such as chromatin-associated protein and high mobility group box 1 (HMGB1) [43]. Some kinds of *S. aureus* have also been found to produce an extracellular, two-component, heterodimeric perforating cytolytic toxin (a leukocidal toxin, PVL), which is extremely virulent, promotes necroptosis and apoptosis, kills human polymorphonuclear cells, and causes severe infections. In necrotizing pneumonia, PVL upregulates the expression of pro-inflammatory cytokines and contributes to the activation of the JAK/STAT pathway, thereby increasing the severity of pneumonia [66]. PVL can also lead to the rapid disruption of mitochondrial homeostasis and apoptosis induced by the activation of cystatase-9 and cystatase-3 [65]. PSMs induce the necroptosis of neutrophils through the activation of mixed-spectrum kinase-like protein (MLKL) phosphorylation and increased release of lactate dehydrogenase, which mediates TNF-α secretion via formyl peptide receptor 2 (FPR2) and leads to the exacerbation of pneumonia [68].

#### 5.2.4. CF

Pulmonary infections with *S. aureus* are highly prevalent in patients with CF, leading to a much higher mortality rate in pulmonary CF. *S. aureus* widely expresses adhesion molecules that recognize, adhere to, and internalize in lung epithelial cells, thereby protect the bacteria from the host immune system and promote chronic infection [20]. Mast cells are activated in infected lung epithelia and release multiple inflammatory cytokines during *S. aureus* infection via the C-KIT (CD117, a tyrosine kinase type III)-activated phosphatidylinositol 3-kinase (PI3K)/AKT/P65 NF-κB signaling pathway [217]. IL-8 expression is increased, and staphylococci entering the lungs are surrounded by a large number of immune cells, macrophages, and a small amount of fibronectin/collagen, causing a strong immune response [295,296]. Intracellular *S. aureus* was found to induce host cell death in epithelial cells via caspase protease [297]. In contrast, macrophages are the main reservoir of *S. aureus* in the body, which not only fail to effectively kill engulfed *S. aureus*, but also induce *S. aureus* to become resistant to a variety of antibiotics [20]. *S. aureus* can form chronic infections through biofilm formation, transformation to small colony variants, the emergence of hypervariable strains, the downregulation of virulence genes, and the formation of heterogeneous bacterial populations [298].

CF is an immunodeficiency disease of autophagy with defective transmembrane conductance regulator (CFTR) genes. The defective gene allows autophagy mediated by ROS. Autophagy inhibits aggregate formation and creates inflammation in the lungs [299,300]. *S. aureus* regulates the expression of ROS, so *S. aureus* infection may have an effect on autophagy in CF [273,274].

### 5.3. Food Poisoning

*S. aureus* is a major cause of food poisoning, which usually occurs after the ingestion of different foods, especially processed meats and dairy products, which are contaminated with *S. aureus* due to improper handling or subsequent improper storage. *S. aureus* is an opportunistic Gram-positive bacterium that actively interacts with mucosal epithelial cells in the intestine, leading to an inflammatory response.

Upon exposure of gastrointestinal epithelial cells to *S. aureus*, epithelial cells express and secrete pro-inflammatory mediators (including IL-8), leading to the activation and attraction of PMN, which recruit multiple immune cells to inflamed intestinal tissues, resulting in intestinal inflammation [23].

LTA provides resistance to host bactericidal factors and promotes adhesion in epithelial cells, contributing to the colonization of *S. aureus* in the gastrointestinal tract [127]. Moreover, LTA can also enhance the inflammatory response of macrophages together with PGN, promote phagosome maturation through activation of the JAK-STAT pathway, and lead to an inflammatory response, leading to multi-organ dysfunction [141]. However, LTA can also lead to the continued accumulation of PMN at the site of inflammation, resulting in chronic inflammation [23].

*Staphylococci* can also induce small intestinal colitis by Rho-inactivated epidermal differentiation inhibitor (EDIN) toxin-mediated enterocytosis, which disrupts the epithelial barrier. Production of β-defensins by the intestinal epithelium contributes to host defense against staphylococcal invasion [301].

In contrast, the δ toxin was found to increase vascular permeability in guinea pig skin and cyclo-AMP levels in the ileum in guinea pig experiments. It also increases the bidirectional movement of Na^+^ and Cl^−^ in the mucosa and inhibits water absorption, thus increasing the severity of diarrhea [75].

The *Staphylococcal* α-toxin also induces a decrease in transmembrane electrical impedance (TER) and a decrease in cellular connexin levels, resulting in a disruption of barrier integrity and a significant decrease in calcium inward flow and TER of intracellular calcium chelation regulation in human intestinal epithelial Caco-2 cells [52]. Elevated intracytoplasmic Ca2^+^ leads to increased mitochondrial Ca2^+^ concentrations and activation of calpain and caspase, resulting in cell lysis [302]. The α-toxin may lead to intestinal epithelial barrier dysfunction and promote the release of intestinal bacteria into the underlying tissues and blood, thus further worsening sepsis [53].

*S. aureus* infections in the intestinal tract expressing superantigenic SEs can cause acute gastroenteritis with a rapid onset of symptoms, causing nausea and severe vomiting, with diarrhea [303]. SEs reach the lamina propria through the cupula and epithelium of the intestinal epithelium and induce the release of 5-hydroxytryptamine (5-HT/5-hydroxytryptamine precursor) from mast cells; 5-hydroxytryptamine can interact with the vagus nerve to induce an emetic response [304]. Enterotoxin SAgs can also penetrate the intestinal lining, act on T cells, macrophages, and monocytes of the immune system, and activate local and systemic immune responses, resulting in significant T cell proliferation, the inhibition of IgM synthesis, and the release of inflammatory mediators (including histamine, leukotrienes, and neurointestinal peptide substance P), which can lead to vomiting, with the patient’s condition becoming more severe as more toxins are added [305,306,307,308]. SEs can also inhibit the reabsorption of water and electrolytes in the small intestine, leading to diarrhea, and the activation of the local immune system can also lead to gastrointestinal damage associated with SE intake [309]. The SEs SEA [94], SEC [150], SEM [99], SEO [97], and SEQ [100] have emetic activities. SEA, SEC, and SEQ also have supercarcinogenic activity. SEB increases numbers of T lymphocytes, concentrations of pro-inflammatory cytokines, and inflammatory mediators in the intestinal mucosa, and increases mucosal permeability and intestinal secretion [97,150,310,311,312]. SEI reversibly disrupts the enterocyte ultrastructure and exhibits weak emetic activity [95,97]. Together with SEH, both can lead to food poisoning.

It was found that microbial colonization of the intestinal epithelium could obtain nutrients by inducing apoptosis in intestinal epithelial cells and that mammalian nutrients released through cystathione-3/7-dependent apoptosis could promote the growth of a variety of Enterobacteriaceae [313]. Whether *S. aureus* infection in the intestinal tract has a similar effect, however, has not been studied.

### 5.4. Autoimmune Diseases

In GPA, the nasal carriage of *S. aureus* is a common trigger for the onset of GPA. In *S. aureus* infection, increased B lymphocyte stimulating factor leads to an increase in B lymphocytes, which secrete ANCA (ANCA is the virulence antibody associated with GPA). ANCA binds to protease 3 on the surface of neutrophils and monocytes, releasing ROS, proteases, cytokines, and neutrophil extracellular trap products (NET-derived products) [221]. Neutrophil extracellular traps (NETs) are extracellular structures composed of chromatin and granule proteins that bind and kill microorganisms which are distinct from necroptosis and apoptosis [314]. Autoreactivity to myeloperoxidase (MPO) can also lead to antineutrophil cytoplasmic antibody (ANCA)-associated vasculitis (AAV) [315].

In hereditary phagocytic dysfunction, which is a CGD, there is an increased susceptibility to bacterial and fungal infections and an increased susceptibility to granuloma formation due to an excessive inflammatory response [316]. Affecting recurrent skin, airways, lymph nodes, liver, brain, and bone, *S. aureus* infections are often life-threatening. Host immune cells undergo ROS-dependent regulation. Phagocytic ROS production is fundamental to the pathogen elimination and regulation of the inflammatory response to infection. The loss of function within the phagocytic NADPH oxidase NOX2 by mutations leads to the impaired production of ROS [273,274,275]. Respiratory bursts attack ROS produced by iron–sulfur cluster-containing proteins, including TCA cycle enzymes, resulting in reduced respiration, reduced ATP, and increased antibiotic tolerance [20]. In neutrophils, the calmodulin-mediated inhibition of calcium-dependent phospholipase A2 enzyme (iPLA2) controls granule and vesicle cytostasis in the phagosome and extracellular microenvironment while correcting defective intracellular and extracellular microbicidal activity. In patients with CGD presenting with defective production of ROS, the increase in non-oxidative killing pathways partially corrected their bactericidal defects [317]. OLFM4 (a neutrophil protein) negatively regulated host defense against bacterial infection and reduced immune defense against *S. aureus* in mice with CGD [24].

### 5.5. Osteomyelitis

In osteomyelitis, SpA binds directly to TNFR-1 expressed on osteoblasts, leading to the activation of NF-κB and translocation to the nucleus of osteoblasts, leading to the release of IL-6 (an important osteoprogenitor), the induction of apoptosis, and the inhibition of the mineralization of osteoblasts. Infected osteoblasts also increase the expression of RANKL (a key protein involved in the initiation of bone resorption), recruiting and activating bone-resorbing cells and osteoclasts, leading to bone loss [117,118]. *S. aureus* also inhibits the expression of alkaline phosphatase, collagen type I, osteocalcin, osteopontin, and osteocalcin, thereby inhibiting bone formation [25].

*S. aureus* also plays a role in nephrapostasis, cancer, bacteremia, pulmonary infections, venous infections, and subcutaneous infections by inducing apoptosis and in sepsis by inducing autophagy [16].

### 5.6. MRSA

MRSA and its antimicrobial resistance exacerbate global health threats, and drug abuse promotes the emergence of bacterial drug resistance. MRSA is the result of a large number of virulence factors produced by *S. aureus* with β-lactam resistance and the resistance of most clones to other antibiotic classes. MRSA has been shown to develop multiple mechanisms of resistance, including cell wall thickening, increased efflux pumps on the cell membrane, mutations in cytoplasmic drug targets, and the modification of drugs and changes in bacterial colonization status [318]. MRSA resistance usually arises through the acquisition of a non-native gene encoding a penicillin-binding protein (PBP2a) with a significantly reduced affinity for β-lactams. This resistance allows cell wall biosynthesis (the target of β-lactams) to continue even in the presence of certain inhibitory concentrations of antibiotics [319]. PBP2a, a PGN transpeptidase, is encoded by the mecA gene, which is carried on a unique mobile genetic element (SCCmec). MecA’s expression is controlled through a protein hydrolysis signaling pathway containing a sensor protein (MecR1) and an inhibitory factor (MecI) that allows *S. aureus* to become resistant to most members of the β-lactam class of antibiotics [320]. FMMs are formed by the interaction of membrane carotenoids with scaffolding proteins, and interference with FMM assembly using existing drugs interferes with PBP2a oligomerization and disables MRSA penicillin resistance in vitro and in vivo, leading to the sensitivity of MRSA infection to penicillin therapy [321]. MRSA infections remain a major global healthcare problem. Of concern is *S. aureus* bacteremia, which has high morbidity and mortality and can cause metastatic or complicated infections, such as infective endocarditis or sepsis. MRSA infections often lead to poorer clinical outcomes [322]. MRSA can promote the degradation of inflammasome components through autophagy and promote their persistence in keratinocytes [271]. Host cells can also prevent MRSA-induced sepsis by attenuating Th1 and Th17 responses through autophagy and by promoting mitochondrial autophagy [323,324]. IL-19, -20, and -24 signals through type I and type II IL-20 receptors inhibit cutaneous IL-1β and IL-17A production to promote MRSA infection in mice [325].

### 5.7. DFI

DFI is a very common and serious complication of diabetes mellitus, with a prevalence of up to 25% in diabetic patients [326]. DFI is caused by diabetes mellitus-induced lower limb vasculopathy, which triggers neuropathy and causes infection of the soft tissue and bone below the ankle joint. DFI can even lead to amputation in severe cases [327]. *S. aureus* is the most common infecting organism in DFI patients, with an infection rate of 42.1%, of which 70% are MRSA [328]. Endogenous flora is the main source of *S. aureus* in DFI patients. The nasal carriage of *S. aureus* is a risk factor for DFI. *S. aureus* infection with nasal carriage was detected in 30.87% of patients with a diabetic foot [329]. Among those patients infected with *S. aureus*, 40.85% were considered MRSA [329].

*S. aureus* is the most virulent pathogen in DFI and induces DFI by expressing a variety of virulence factors [330]. The detection of virulence factors Cap8, Sea, Sei, LukDE (leukocidin), and hlgv (haemolysin) allows for the grading of grade 1 and grade 2–4 ulcers. Toxicity factor testing can also predict wound outcomes. Therefore, the detection of virulence factors provides a useful basis for the treatment of diabetic foot ulcers [331]. Among these genes, SEA has been shown to have a strong pro-inflammatory effect [94]. The virulence factor epidermal cell differentiation inhibitor (EDIN), which can affect epithelial and endothelial cohesion by inhibiting RhoA, facilitates the spread of *S. aureus* to infect deeper tissues [332]. It was found that 71.4% of EDIN-positive DFI patients had moderate to severe infections and that 28.6% had mild infections [333]. Different strains of *S. aureus* pathogenic bacteria can produce three subtypes of EDIN, such as EDIN type A (EDIN-A), EDIN type B (EDIN-B), or EDIN type C (EDIN-C). The majority of EDIN-positive strains were found to be EDIN-B positive (86.7%), suggesting that EDIN-B is a co-predictive risk marker for DFI [333].

Treatment of DFI can begin with debridement of the wound, followed by irrigation and disinfection of the wound with an antiseptic and the use of antibiotics. The choice of antibiotics varies depending on the severity of the infected wound of DFI patients. Antibiotic treatment can be either oral or intravenous. Clindamycin is preferred for mild cases [334]. Gentamicin can be used to treat moderate and severe cases [335]. Vancomycin can be used for more severe DFI patients and can be used to combat MRSA infection [336].

MRSA is a drug-resistant organism of *S. aureus* and is highly resistant to most β-lactam antibiotics, as well as to a wide range of other anti-microbials. This makes MRSA infections difficult and costly to treat. MRSA can form a biofilm in vitro that is resistant to antibiotics, and biofilm formation is one of the main determinants of the prevalence of pathogens associated with medical device infections. Therefore, in the treatment of DFI, the inhibition of in vitro MRSA biofilm formation often requires high doses of broad-spectrum antibiotics in combination with multiple drugs. Ceflorin and gentamicin have also been found to have an inhibitory effect on biofilms isolated from diabetic feet [337].

Antimicrobial peptides (AMPs) are an adjunct to conventional therapies [338]. Nisin, a type of AMP, is produced by lactic acid bacteria. Nisin can inhibit *S. aureus* and other Gram-positive bacteria effectively. To prolong the half-life of Nisin in vivo, guar gum biogel has been used as a delivery agent. Nisin biogel was able to inhibit the formation of the biofilm of *S. aureus* isolated from DFI [339].

Chlorhexidine is widely used as a skin antiseptic and is one of the most commonly used disinfectants in the treatment of DFI. Furthermore, chlorhexidine can be used to inhibit *S. aureus* and MRSA infection [340]. In addition, chlorhexidine inhibits the formation of biofilms [341]. It was found that the combination of chlorhexidine and AMP niosomal-biogel was more efficient in inhibiting biofilm formation than either of them alone [342].

Another treatment option for alternative antibacterial drugs is the use of natural compounds from plants that have antibacterial properties. *Urtica dioica* and *Lavandula angustifolia* extracts were found to have an inhibitory effect on MRSA isolated from diabetic feet, suggesting that topical applications of lavender and nettle leaves could be used to treat DFI [343].

As there is no effective treatment for DFI, it is important to study the mechanisms of immune cells, cell death in *S. aureus*, and virulence factors, especially MRSA, for the treatment of DFI.

A summary of the diseases caused by *S. aureus* is shown in Figure 3.

## 6. Conclusions and Future Perspectives

In this review, we have summarized the virulence factors produced by *S. aureus*, the types of immune cells, the cell death mechanisms activated by *S. aureus*, and the pathogenesis of *S. aureus* in several diseases. A better understanding of these may contribute to targeted therapies for *S. aureus*-induced diseases in the future.

Two areas of investigation remain open for major advancement. First, the mechanisms of Th9, Tfh, and ILC cells activated by *S. aureus* and virulence factors remain largely unknown. Second, the roles of ferroptosis and copper-induced cell death in diseases caused by *S. aureus* remain poorly defined. These may become new directions for future research.

## Figures and Tables

**Figure 1 toxins-14-00464-f001:**
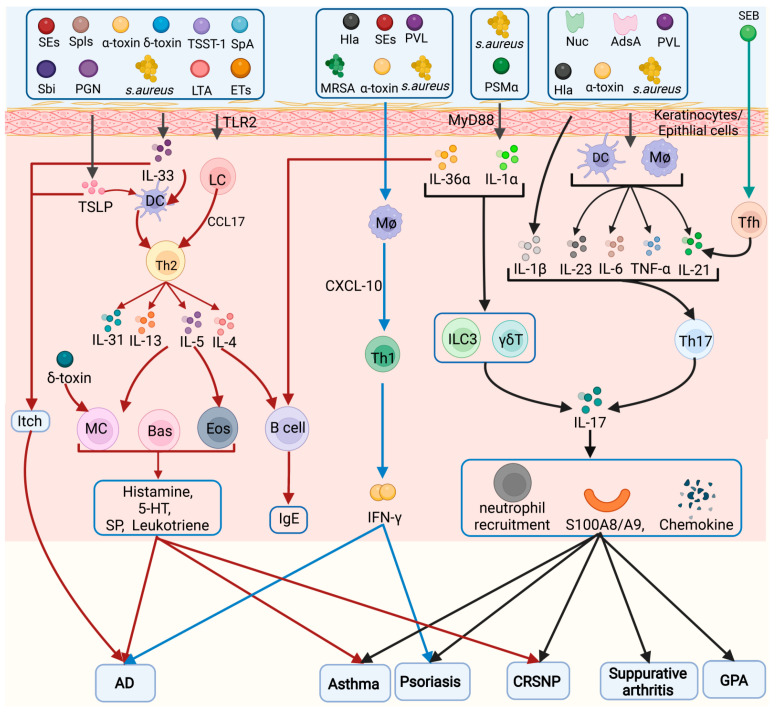
Inflammatory cell types in the pathogenesis of *Staphylococcus aureus*. Different virulence factors of *S. aureus* can induce activation of Tfh, Th1, Th2, Th9, and Th17 cells, which play a role in chronic sinusitis, AD, asthma, itch, psoriasis, septic arthritis, and CGD. Eos: eosinophils, Bas: basophils, MC: mast cells, Mø: macrophage.

**Figure 2 toxins-14-00464-f002:**
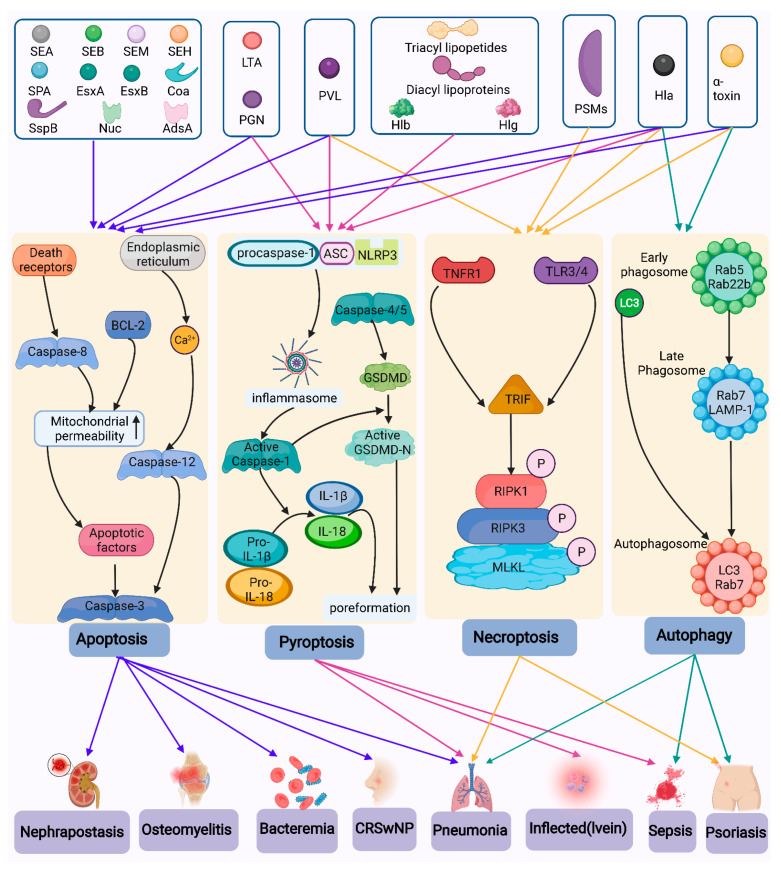
Types of cell death induced by *S. aureus.* The multiple toxins of *S. aureus* can induce multiple modes of cell death, including apoptosis, necroptosis, pyroptosis, and cell autophagy, and play a role in diseases as diverse as renal abscesses, septic arthritis, cancer, bacteremia, chronic sinusitis, pneumonia, venous infections, sepsis, psoriasis, and diseases caused by MRSA. Blue arrows represent apoptotic pathways; purple arrows represent the pyroptosis cell death pathway; yellow arrows represent the necroptotic pathway; green arrows represent the autophagic pathway.

**Figure 3 toxins-14-00464-f003:**
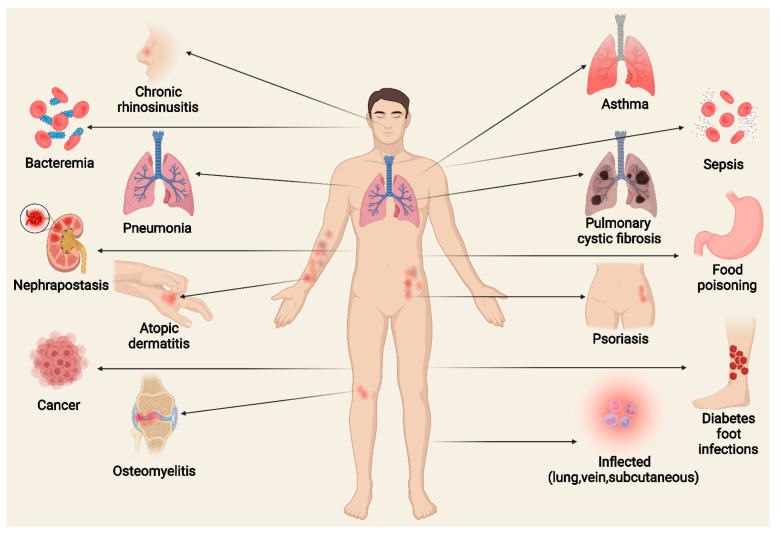
Types of diseases caused by *S. aureus*. *S. aureus* can cause a variety of systemic diseases, including skin diseases, respiratory diseases, food poisoning, autoimmune diseases, osteomyelitis, DFI, and the formation of MRSA.

**Table 1 toxins-14-00464-t001:** List of abbreviations and their full forms used in this review.

Abbreviation	Full Name	Abbreviation	Full Name
5-HT	5-hydroxytryptamine	MMPs	Matrix metal proteinases
AAV	Associated vasculitis	MPO	Myeloperoxidase
	Atopic dermatitis	MRSA	Methicillin-resistant *S. aureus*
AdsA	Adenosine synthase A	MSCRAM	Mucin matrix molecules
Agr	Auxotrophic gene regulator	MSCRAMM	Microbial surface components recognizing adhesive matrix molecules
AMP	Antimicrobial peptide	NETs	Neutrophil extracellular traps
AMs	Alveolar macrophages	NF-κB	Nuclear factor kappa B
ANCA	Antineutrophil cytoplasmic antibody	NK	Natural killer cells
BCL-6	B-cell lymphoma-6	NO	Nitric oxide
BMECs	Bovine mammary epithelial cells	Nuc	Nuclease
CF	Cystic fibrosis	PA	Protein A
CFTR	Conductance regulator	PAMPs	Pathogen-associated molecular patterns
CGD	Chronic granulomatous disease	PBP2a	Penicillin-binding protein
Coa	Coagulase	PCD	Programmed cell death
CpG	Cytosine–phosphate–guanine	PD-1	Programmed death 1
CRSwNP	Chronic rhinosinusitis with nasal polyposis	PFTs	Pore-forming toxins
CWA	Cell wall-anchored	PGN	Peptidoglycan
CXCR5	CXC chemokine receptor 5	PI3K	Phosphatidylinositol 3-kinase
DAMPs	Damage-associated molecular patterns	PIP	Phosphatidylinositol phosphate
DCs	Dendritic cells	PMN	Polymorphonuclear leukocytes
DFI	Diabetic foot infection	PRRs	Pattern recognition receptors
Dsg1	Desmoglein 1	PSMs	Phenol-soluble modulins
Eap	Extracellular adhesion protein	PVL	Panton–Valentine leukocidin
ECP	Eosinophil cationic protein	RIPK1	Receptor-interacting protein kinase 1
EDIN	Epidermal differentiation inhibitor	ROS	Reactive oxygen specie
eDNA	Extracellular DNA	*S. aureus*	*Staphylococcus aureus*
EPO	Eosinophil peroxidase	SAgs	Superantigens
ESS	Early-secretion antigen-6 secretion system	Sbi	Second immunoglobulin-binding protein
ETs	Exfoliative toxins	SEs	Staphyloccocal enterotoxins
FPR2	Formylpeptide receptor 2	Socs3	Suppressor of cytokine signaling 3
GC	Germinal center	SpA	*Staphylococcal* protein A
G-CSF	Granulocyte colony-stimulating factor	Spls	Serine protease-like proteins
GM-CSF	Granulocyte-macrophage colony-stimulating factor	SspB	Staphopain B
GPA	Granulomatous polyangiitis	SSSS	Staphylococcal scalded skin syndrome
Hla	α-hemolysin	TCA	Tricarboxylic acid
Hlb	β-hemolysin	Tem	Memory T
Hlg	γ-hemolysin	TER	Transmembrane electrical
I-CAM	Intercellular adhesion molecules	Tfh	Follicular helper T
IFN-γ	Interferon γ	TGF-β	Transforming growth factor β
lgE	Immunoglobulin-E	Th	T helper
IL	Interleukin	TLR	Toll-like receptor
ILCs	Innate lymphoid cells	TNF	Tumor necrosis factor
iPLA2	Inhibition of calcium-dependent phospholipase A2	TNFR1	Tumor necrosis factor receptor 1
JNK	Jun N-terminal kinase	Treg	Regulatory T
LCs	Langerhans cells	TSST-1	Toxic shock syndrome toxin 1
LTA	Lipoteichoic acid	V-CAM	Vascular cell adhesion molecules
MDP	Muramyl dipeptide	VEGF	Vascular endothelial growth factor
MLKL	Mixed-spectrum kinase-like protein		

**Table 2 toxins-14-00464-t002:** The mechanisms and characteristics of different types of SEs.

SEs	Functions
SEA	Anti-protein hydrolase, supercarcinogenic and emetic toxin, causes food poisoning, but no diarrheal activity
SEB	Increase in the number of T lymphocytes, as well as the concentration of pro-inflammatory cytokines and inflammatory mediators in the intestinal mucosa; increase mucosal permeability and intestinal secretion
SEC	Supercarcinogenic and emetic toxin, but no diarrheal activity
SEG	Reversible destruction of intestinal cell ultrastructure
SEH	Causes food poisoning
SEI	Reversibly disrupts enterocyte ultrastructure and exhibits weak emetic activity
SEM	Strong emetic potential
SEO	Has emetic activity
SEQ	Significantly stable for heat treatment and digestive enzyme degradation, and exhibits significant supercarcinogenic and emetic activity

**Table 3 toxins-14-00464-t003:** Inflammatory cell types and cell death mechanisms involved in different *Staphylococcus aureus* virulence factors.

Classification		Virulence Factors	Cell Types	Cell Death Mechanisms	Diseases	Refs.
Secreted Virulence factors	PFT	Hla	Keratinocytes Macrophages	Pyroptosis Necroptosis Autophagy Apoptosis		[43,143,144,145,146]
		Hlb		Pyroptosis		[34]
		Hlg		Pyroptosis		[35]
		α-Toxin	Th1, Th2 Neutrophils Macrophages	Apoptosis Necroptosis Autophagy	AD GPA Pneumonia Chronic sinusitis sepsis	[47,48,49,50,51,52,53]
		PVL	DCs Macrophages	Apoptosis	Pneumonia	[58,62,63,64,65,66]
	PSMs	PSMα	Keratinocytes Neutrophils	Necroptosis	AD Pneumonia	[5,67,68,69,70]
		PSMβ	Neutrophil			[73]
		PSMγ (δ toxin)	Mast cells Keratinocytes		AD Diarrhea	[74,75,147]
	SAgs	SEA	B cells CD4^+^ T cells Keratinocytes Eosinophils	Apoptosis	AD Asthma Food poisoning	[13,91,92,148,149]
		SEB	Th1, Th2 Eosinophils Proximal renal tubular epithelial cells	Apoptosis	Chronic sinusitis AD Asthma	[15,92,93,149]
		SEC	B cells		AD Food poisoning	[13,150]
		SEG			Food poisoning	[95]
		SEH	Bovine mammary gland epithelial cells	Apoptosis	Food poisoning	[96]
		SEI			Food poisoning	[95,97]
		SEM		Apoptosis	Food poisoning	[98,99]
		SEO			Food poisoning	[97]
		SEQ			Food poisoning	[100]
		TSST-1	B cells Keratinocytes		AD	[13,50]
	Proteases	ETs			AD	[31]
		Spls	T cells		Asthma SSSS	[78] [76]
		SspB	Macrophages	Apoptosis		[85]
	Secreted Enzymes (proteins)	EsxA, EsxB		Apoptosis		[101,102]
		Coa		Apoptosis		[104]
		Eap	T cells Lung epithelial cells		Psoriasis CF	[109]
		Nuc	Neutrophils Macrophages	Apoptosis	Bacteraemia Nephrapostasis	[108]
		AdsA	Neutrophils Macrophages	Apoptosis	Bacteraemia Nephrapostasis	[107]
Cell wall components	Surface Proteins (CWA)	Sbi	Keratinocytes		AD	[119]
		SpA	Tumor cells	Apoptosis		[115,116,117,118]
	PAMPs	Triacyl lipopetides		Pyroptosis	Sepsis	[69,124]
		Diacyl lipoproteins		Pyroptosis	Sepsis	[69,151]
		LTA	Neutrophils Macrophages B cell	Pyroptosis	Food poisoning	[23,125,126,127]
		PGN	LCs Mast cells Macrophages Macrophages	Autophagy	AD Sepsis Pneumonia Psoriasis	[17,130,131,132,133,140,141]

Note: The virulence factors of *S. aureus*, i.e., PFTs, PSMs, ETs, SAgs, enzymes, effectors, PAMPs, proteins, and peptides, induce different kinds of cellular activation and different forms of cell death in different diseases.

## Data Availability

Not applicable.

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
