# Peer review of "Exploring the Role of Staphylococcus aureus in Inflammatory Diseases"

_toxins, 2022, doi:10.3390/toxins14070464_

Round 1
Reviewer 1 Report
This manuscript should be checked in detail by authors.

Author Response
Response to Reviewer 1 Comments
This manuscript is well included about S. aureus and inflammation and inflammatory
diseases. However, usage of abbreviated words is not unified.
Comments:
1. Phenol-soluble modulins (PSMs, line 46) and phenol-soluble regulatory protein (PSMs, line 235) are different molecule? If so, abbreviations should be distinguished.
Response: Thanks for your suggestion. Phenol-soluble modulins and phenol-soluble regulatory protein are the same molecules. We have revised this mistake and used “phenol-soluble modulin” for the first mention in line 75, and the abbreviation “PSM” is used for all subsequent appearances.
2. The authors used both exfoliating toxins (line 64) and exfoliative toxins (line 136). Exfoliative toxin is a prefer name.
Response: Thanks for your suggestion. We used “exfoliative toxins” for the first mention in line 75, and the abbreviation “ETs” is used for all subsequent appearances.
3. TNF-b should be described as tumor necrosis factor b, not b tumor necrosis factor at line 258.
Response: Thanks for your good suggestion. We followed the suggestion, and changed “b tumor necrosis factor” to “tumor necrosis factor b” in line 341.
4. Lipoteichoic acid (LTA line 190), lipophosphatidic acid (LTA, line 497), and lipophlin acid (LTA, line 545) are different molecules? If so, abbreviations should be distinguished.
Response: Thanks for your suggestion. Lipoteichoic acid, lipophosphatidic acid, and lipophlin acid are the same molecules. We have revised these mistakes and used “Lipoteichoic Acid” for the first mention in line 270, and the abbreviation “LTA” is used for all subsequent appearances.
5. Innate lymphocytes are not correct (line 231). We usually call them as “innate lymphoid cells (ILCs).
Response: Thanks for your good suggestion. We followed the suggestion, and changed “Innate lymphocytes” to “innate lymphoid cells (ILCs)” in line 37. We have revised these mistakes and used “innate lymphoid cells” for the first mention in line 39, and the abbreviation “ILCs” is used for all subsequent appearances.
b transforming growth factor beta is incorrect. First “b” should be delated.
Response: Thanks for your good suggestion. We followed the suggestion, and delated the first “b” in line 395.
Abbreviated word should be used when once words were abbreviated.
1) Atopic dermatitis (AD) line 46, lines 99, 102, 120, 128, 131, 148, 149, 157, 213, 215, 220, 237, 264, 278, 361, 401, 403, 436, 459, 470, 514, 539, 685, 796, 797, 804, 805, 807, 808, 817, 822, 823, 843,
Response: Thanks for your suggestion. We used “Atopic dermatitis (AD)” for the first mention in line 50, and the abbreviation “AD” is used for all subsequent appearances.
2) Cystic fibrosis should be defined as CF at line 46, lines 227, 952, 954, 970, 974
Response: Thanks for your suggestion. We used “cystic fibrosis (CF)” for the first mention in line 50, and the abbreviation “CF” is used for all subsequent appearances.
3) methicillin-resistant S. aureus (MRSA) should be defined at line 52, line 766, 1080
Response: Thanks for your suggestion. We used “methicillin-resistant S. aureus (MRSA)” for the first mention in line 48, and the abbreviation “MRSA” is used for all subsequent appearances.
4) Pore-forming toxins (PFTs) line 63, lines 67, 68, 70, 76, 106, 585
Response: Thanks for your suggestion. We used “Pore-forming toxins (PFTs) for the first mention in line 74, and the abbreviation “PFTs” is used for all subsequent appearances.
5) Phenol-soluble modulins (PSMs) line 64, lines 113, 114, 126, (235, 238), 661, 820, 829
Response: Thanks for your suggestion. We used “phenol-soluble modulin” for the first mention in line 75, and the abbreviation “PSM” is used for all subsequent appearances in lines.
6) Superantigens (sAgs) line 64-65, lines 136, 137, 473, 534, 1015
Response: Thanks for your suggestion. We used “Superantigens (sAgs)” for the first mention in line 75, and the abbreviation “sAgs” is used for all subsequent appearances.
7) Exfoliating/exfoliative toxinxs (Ets) line 64, lines 129, 130, 821, 1144
Response: Thanks for your suggestion. We used “exfoliative toxins (ETs)” for the first mention in line 75, and the abbreviation “ETs” is used for all subsequent appearances.
8) a-hemolysin (Hla) line 73, lines 75, 80-81, 82, 627-628, 661, 696, 743, 748, 906, 935, 936
Response: Thanks for your suggestion. We used “a-hemolysin (Hla)” for the first mention in line 84, and the abbreviation “Hla” is used for all subsequent appearances.
9) Panton-Valentine leucocidin (PVL) line 74, lines 105, 106, 627, 694, 703
Response: Thanks for your suggestion. We used “Panton-Valentine leucocidin (PVL)” for the first mention in line 85, and the abbreviation “PVL” is used for all subsequent appearances.
10) Interleukin should be defined as IL at line 84, lines 257, 309, 574, 589, 634, 733, 734, 1070, 1107
Response: Thanks for your suggestion. We used “Interleukin(IL)” for the first mention in line 94, and the abbreviation “IL” is used for all subsequent appearances.
11) Nitric oxide should be defined as NO at line 86, lines 335, 715
Response: Thanks for your suggestion. We used “Nitric oxide (NO)” for the first mention in line 97, and the abbreviation “NO” is used for all subsequent appearances.
12) Tumor necrosis factor receptor 1 should be defined as TNFR1 at line 109, line 653
Response: Thanks for your suggestion. We used “Tumor necrosis factor receptor 1 (TNFR1)” for the first mention in line 121, and the abbreviation “TNFR1” is used for all subsequent appearances.
13) Serine protease-like proteins (Spls) line 131, line 905
Response: Thanks for your suggestion. We used “Serine protease-like proteins (Spls)” for the first mention in line 16, and the abbreviation “Spls” is used for all subsequent appearances.
14) Tumor necrosis factor should be defined as TNF at line 142, line 669
Response: Thanks for your suggestion. We used “Tumor necrosis factor (TNF)” for the first mention in line 187, and the abbreviation “TNF” is used for all subsequent appearances.
15) Staphylococcal enterotoxins (SEs) line 142, lines 263, 474, 516, 598, 601, 707, 821, 839, 843, 847, 850-851, 885, 891, 923, 1009, 1011, 1015, 1020, 1023
Response: Thanks for your suggestion. We used “Staphylococcal enterotoxins (SEs)” for the first mention in line 188, and the abbreviation “SEs” is used for all subsequent appearances.
16) Toxic shock syndrome toxin-1 (TSST-1) line 143, lines 154, 155, 820
Response: Thanks for your suggestion. We used “Toxic shock syndrome toxin-1 (TSST-1)” for the first mention in line 189, and the abbreviation “TSST-1” is used for all subsequent appearances.
17) Coagulase (Coa) line 161, lines 168, 169, 695
Response: Thanks for your suggestion. We used “Coagulase (Coa)” for the first mention in line 211, and the abbreviation “Coa” is used for all subsequent appearances.
18) Nuclease (Nuc) line 161, lines 173, 695, 726
Response: Thanks for your suggestion. We used “Nuclease (Nuc)” for the first mention in line 211, and the abbreviation “Nuc” is used for all subsequent appearances.
19) Adenosine synthase A (AdsA) line 161, lines 696, 726
Response: Thanks for your suggestion. We used “Adenosine synthase A (AdsA)” for the first mention in line 211, and the abbreviation “AdsA” is used for all subsequent appearances.
20) Staphopain B (SspB) line 184, lines 185, 696, 729, line 696 staphoain B is incorrect.
Response: Thanks for your suggestion. We used “Staphopain B (SspB)” for the first mention in line 176, and the abbreviation “SspB” is used for all subsequent appearances.
21) Pathogen-associated molecular patterns (PAMPS) line 189, line 577
Response: Thanks for your suggestion. We used “Pathogen-associated molecular patterns (PAMPS)” for the first mention in line 67, and the abbreviation “PAMPS” is used for all subsequent appearances.
22) Lipoteichoic/lipophosphatidic/lypophilin acid (LTA) lines 190, 497, 545, lines 201, 202, 625/985, 987, 990
Response: Thanks for your suggestion. We used “Lipoteichoic acid (LTA)” for the first mention in line 270, and the abbreviation “LTA” is used for all subsequent appearances.
23) Toll-like receptor (TLR) line 196, lines 196-197, 198, 253, 448, 636, 670, 809, 815, 914
Response: Thanks for your suggestion. We used “Toll-like receptor (TLR)” for the first mention in line 99, and the abbreviation “TLR” is used for all subsequent appearances.
24) Second immunoglobulin-binding protein (Sbi) line 209, lines 212, 245, 836
Response: Thanks for your suggestion. We used “Second immunoglobulin-binding protein (Sbi)” for the first mention in line 268, and the abbreviation “Sbi” is used for all subsequent appearances.
25) Peptidoglycan (PGN) line 209, lines 216, 217, 282, 288, 463, 626, 638, 639, 762, 988, 1091
Response: Thanks for your suggestion. We used “Peptidoglycan (PGN)” for the first mention in line 245, and the abbreviation “PGN” is used for all subsequent appearances.
26) Langerhans cells should be defined as LC at line 217, lines 283, 288,359,443,446
Response: Thanks for your suggestion. We used “Langerhans cells(LCs)” for the first mention in line 303, and the abbreviation “LCs” is used for all subsequent appearances.
27) Innate lymphoid cells (ILCs) line 231, lines 394, 395, 900
Response: Thanks for your suggestion. We used “Innate lymphoid cells (ILCs)” for the first mention in line 39, and the abbreviation “ILCs” is used for all subsequent appearances.
28) Regulatory T (Treg) line 287, lines 355, 550
Response: Thanks for your suggestion. We used “Regulatory T (Treg)” for the first mention in line 372, and the abbreviation “Treg” is used for all subsequent appearances.
29) Transforming growth factor beta (TGF-beta) line 310, line 485
Response: Thanks for your suggestion. We used “Transforming growth factor β (TGF-β)” for the first mention in line 395, and the abbreviation “Treg” is used for all subsequent appearances.
30) Matrix metal proteinases (MMPs) line 335, lines 336, 339
matrix metalloproteinases (MMPs)
Response: Thanks for your suggestion. We used “Matrix metal proteinases (MMPs)” for the first mention in line 407, and the abbreviation “MMPs” is used for all subsequent appearances.
31) Interferon (IFN) line 257, lines 311, 462
Response: Thanks for your suggestion. We used “Interferon (IFN)” for the first mention in line 111, and the abbreviation “IFN” is used for all subsequent appearances.
32) Granulocyte colony-stimulating factor (G-CSF) line 346, line 492
Response: Thanks for your suggestion. We used “Granulocyte colony-stimulating factor (G-CSF)” for the first mention in line 431, and the abbreviation “G-CSF” is used for all subsequent appearances.
33) Follicular helper T should be defined as Tfh at line 327, line 365
Response: Thanks for your suggestion. We used “Follicular helper T (Tfh)” for the first mention in line 412, and the abbreviation “Tfh” is used for all subsequent appearances.
34) Granulocyte-macrophage colony-stimulating factor (GM-CSF) line 414, line 524
Response: Thanks for your suggestion. We used “Granulocyte-macrophage colony-stimulating factor (GM-CSF)” for the first mention in line 405, and the abbreviation “GM-CSF” is used for all subsequent appearances.
35) Dendritic cells (DCs) line 424, lines 436, 441, 444, 896
Response: Thanks for your suggestion. We used “Dendritic cells (DCs)” for the first mention in line 40, and the abbreviation “DCs” is used for all subsequent appearances.
36) Polymorphonuclear leukocytes (PMN) line 499, lines 523, 777, 793, 941, 948, 951, 957,
Response: Thanks for your suggestion. We used “Polymorphonuclear leukocytes (PMN)” for the first mention in line 584 and the abbreviation “PMN” is used for all subsequent appearances.
37) Rhinosinusitis without nasal polyps (CRSwNP), line 922
Response: Thanks for your suggestion. We used “Rhinosinusitis without nasal polyps (CRSwNP)” for the first mention in line 458 and the abbreviation “CRSwNP” is used for all subsequent appearances.

Reviewer 2 Report
Dear Authors. The submitted manuscript "Exploring the role of Staphylococcus aureus in inflammatory diseases" provides comprehensive information on the role of various mechanisms associated with S. aureus in human inflammatory response, development of various diseases, etc. In my opinion, the paper is well written and clearly highlights the most important virulence factors. I only have a minor suggestion - as we have seen the increasing problem of diabetes in recent years, and thus diabetes-related disorders are becoming critically important to maintain the proper functioning of the health care system, it would be beneficial to add a short paragraph describing the role of S. aureus in the development of diabetic foot infections. S. aureus is known to be a major causative agent of DFIs in developed countries. In Figure 3 describing the types of diseases caused by S. aureus, DFI should also be added.
Author Response
Response to Reviewer 2 Comments
Response: Thanks for your great suggestions. We followed the suggestions, and added a section “5.7 DFI” in lines 1193-1251 describing the roles of S. aureus and the virulence factors in the development of diabetic foot infections (DFI) and the treatment strategies for DFI caused by MRSA. And we also added DFI intothe revised Figure 3.
Reviewer 3 Report
Be careful at abbreviations, once you define them, use the abbreviation in text. Use a abbreviation list also.
The author use Staphylococcus aureus and then S. aureus, be consecvent.
Introduction is academic (like in a Microbiology class) not scientific, it is in contrast with the rest of the article which is well written, it should be rewritten
Figures 1,2,3 are reproduction or are drawned by the authors ?
Great Reference collection but verify if all references are cited in text. Also verify the style of References, I found some discrepancies at a first overlook
A short conclussion would be useful
There are some langauge, syntax and grammar mistakes in the text
Good review work, can be published in Toxins after a careful text check
Author Response
1)Be careful at abbreviations, once you define them, use the abbreviation in text. Use a abbreviation list also. The author use Staphylococcus aureus and then S. aureus, be consecvent.
Response: Thanks for your helpful suggestions. We followed the suggestions, used “full name (abbreviation)” for the first mention, and uesd the abbreviation for all subsequent appearances and added a abbreviation list in revised Table 1.
2)Introduction is academic (like in a Microbiology class) not scientific, it is in contrast with the rest of the article which is well written, it should be rewritten
Response: Thanks for your good suggestion. We revised the introduction and deleted some sentences which are not scientific in line 22-26, 45-47, 45-56
3)Figures 1,2,3 are reproduction or are drawned by the authors ?
Response: The three figures are designed by the corresponding author and drawned by the first-author.
4)Great Reference collection but verify if all references are cited in text. Also verify the style of References, I found some discrepancies at a first overlook
Response: Thanks for your reminding. We followed the suggestion, and verified all references are cited in text and revised some references which were missing page numbers.
5)A short conclussion would be useful
Response: Thanks for helpful suggestion. We followed the suggestion, and rewrited the “Conclussion and and Future Perspectives” in lines 1253-1261.
6)There are some langauge, syntax and grammar mistakes in the text
Response: Thanks for your suggestion. We followed the suggestion, and corrected some mistakes. And the manuscript has undergone English language editing by MDPI (https://www.mdpi.com/authors/english). The manuscript (English Editing ID: 46107) has been checked for correct use of grammar and common technical terms by native English speaking editors.
Good review work, can be published in Toxins after a careful text check
Response: Thanks for your great comments.

Reviewer 4 Report
This is an important paper and should be published after revision and correction by the authors. It seems to be a new way of communication between different disciplines.
The figures in this part prove clear expertise on the subject of programmed cell death and its forms with apoptosis, necroptosis, inflammasomes and autophagy.
In the second part the author provide a thorough and comprehensive review with very extensive references in the application of medical science, infectious clinical disease, microbial infections, inflammatory diseases and autoimmune conditions, as well as microbiology and biology. This type of information is hopefully also a future trend in medical/biological sciences cooperating between experts in the use of defined technologies between broad experts and specialists in different diseases.
This is a broad and comprehensive review describing many different cell types involved in infections, allergic reactions and the inflammatory process. It would help if the authors could ask for major editing of the area which is not their expertise in detail. The broad understanding and application of the results of the test in details of medical and biological science areas could be a well-used way of communication for the future.
Especially chapter 2 (Virulence factors of S. aureus) needs editing to be readable and comprehensible. In its present form it’s not clearly understandable and has to be corrected. Spelling mistakes as well as grammatical and vocabulary errors affect the readability of this work negatively. For those reasons it should be corrected, but the paper should not be rejected altogether as it is extensively researched and provides a good overview.
Author Response
This is an important paper and should be published after revision and correction by the authors. It seems to be a new way of communication between different disciplines.
The figures in this part prove clear expertise on the subject of programmed cell death and its forms with apoptosis, necroptosis, inflammasomes and autophagy.
In the second part the author provide a thorough and comprehensive review with very extensive references in the application of medical science, infectious clinical disease, microbial infections, inflammatory diseases and autoimmune conditions, as well as microbiology and biology. This type of information is hopefully also a future trend in medical/biological sciences cooperating between experts in the use of defined technologies between broad experts and specialists in different diseases.
This is a broad and comprehensive review describing many different cell types involved in infections, allergic reactions and the inflammatory process. It would help if the authors could ask for major editing of the area which is not their expertise in detail. The broad understanding and application of the results of the test in details of medical and biological science areas could be a well-used way of communication for the future.
Especially chapter 2 (Virulence factors of S. aureus) needs editing to be readable and comprehensible. In its present form it’s not clearly understandable and has to be corrected.
Response: Thanks for your helpful suggestion. We followed this suggestion, we asked several experts from different areas and students to read the manuscript and receive and give some comments from them. And we try to make the revised manuscript readable and comprehensible.
In order to rewrite chapter 2 (Virulence factors of S. aureus) better, we readed a lot of literatures and have a clearer and deeper understanding of virulence factors of Staphylococcus aureus. The changes in the revised manuscript are as follows.
1) In chapter 2, we added the classification of the virulence factors produced by S. aureus, such as a) secreted virulence factors including toxins and superantigens, b) extracellular enzymes, c) surface proteins, and d) pathogen-associated molecular patterns. And we also summarizes the function of each type of virulence factor (in lines 61-68).
2) In each sections (such as 2.1, 2.2, 2.3 et al.), we further classified the subtypes of each virulence factor and described the mechanism enable the pathogen's escape from protective immune responses. (section 2.1.5 “Secreted Enzymes” in lines 204-212, section 2.2 “Surface Proteins of S. aureus” in line 239-252, section 2.2.2 “Staphylococcal protein A” in line 263-267, section 2.3 “PAMPs” in 277-280).
3) Surface Proteins, known as Cell Wall-Anchored (CWA) proteins, play important roles in adhesion, host cell invasion, and immune response evasion. We added section 2.2 named “Surface Proteins of S. aureus” (in lines 238-272).
4) In order to make the revised manuscript readable and comprehensible, we have rewritten most of the virulence factors, such as Hlb and Hlg (line101-104), PVL(line118-122), PSMα (lines 136-139), PSMβ (lines 141-146), PSMγ (δ toxin) (lines 152-156), ETs (lines 162-165), Spls (lines 169-174), SspB (lines 178-181), SAgs (lines 187-202), Nuc and AdsA (lines 223-230), PGN (lines 300-311).
5) We modified Table2 according to revised classification of chapter 2.
Spelling mistakes as well as grammatical and vocabulary errors affect the readability of this work negatively.
Response: Thanks for your suggestion. We followed this suggestion, and the manuscript has undergone English language editing by MDPI (https://www.mdpi.com/authors/english). The manuscript (English Editing ID: 46107) has been checked for correct use of grammar and common technical terms by native English speaking editors.
For those reasons it should be corrected, but the paper should not be rejected altogether as it is extensively researched and provides a good overview.
Response: Thanks for your great comments.

Round 2
Reviewer 3 Report
The article can be published in the current form in Toxins
Reviewer 4 Report
Thank you for carefully taking up my suggestions.